



# The influences of historic lake trophy and mixing regime changes on long-term phosphorus fractions retention in sediments of deep, eutrophic lakes: a case study from Lake Burgäschi, Switzerland

Luyao Tu[1], Paul Zander[1], Sönke Szidat[2], Ronald Lloren[3,4], Martin Grosjean[1]

Oeschger Centre for Climate Change Research and Institute of Geography, University of Bern, Switzerland
Oeschger Centre for Climate Change Research and Department of Chemistry and Biochemistry, University of Bern, Switzerland
Department of Earth Science, ETH Zürich, Switzerland
Eawag, Swiss Federal Institute of Aquatic Science and Technology, Switzerland

*Correspondence to*: Luyao Tu (luyao.tu@giub.unibe.ch)

**Abstract.** Hypolimnetic anoxia in eutrophic lakes can delay lake recovery to lower trophic states via the release of sediment phosphorus (P) to surface waters on short time scales. However, the effects of hypolimnetic redox conditions and eutrophication on long-term sediment P-fraction retention are not clear yet. In this study, we investigated the sediment profiles since the early 1900s from Lake Burgäschi, a deep, eutrophic lake on the Swiss Plateau. The changes of sediment P-fraction retention were assessed with respect to lake trophic evolution (sedimentary green-pigments proxy), hypolimnetic oxygenation regime (Fe/Mn ratio proxy), sediment geochemical characteristics, and lake restoration history. Results showed that long-term retention of total P and labile P-fractions in sediments was predominantly affected by autochthonous Fe and Mn preserved in anoxic sediments, which were controlled by past hypolimnetic redox conditions. By contrast, refractory HCl-P (Ca-P) fraction retention largely resulted from authigenic $CaCO_3$-P precipitation and increased with higher eutrophic levels. The retention of total P and labile P fractions was considerably reduced in surface sediments from 1977-2017, when Lake Burgäschi had the highest eutrophic levels and a persistent anoxic hypolimnion. We attributed the phenomenon to reduced sediment P-binding capacity (Mn and Fe oxyhydroxides) under the eutrophication-induced anoxic hypolimnion and decreased water-P concentrations due to hypolimnetic withdrawal. Our study implies that in seasonally stratified deep lakes like Lake Burgäschi, hypolimnetic withdrawal of P-enriched water can effectively reduce P retention in sediments and potentials of sediment-P release (seen from low P availability after 1977). However, the restoration has not improved lake trophic state, similarly to the findings from lake limnological survey.

**Keywords:** Phosphorus fractions, eutrophication, hypolimnetic anoxia, hypolimnetic withdrawal, deep lakes



 **1 Introduction**

Phosphorus (P) eutrophication in freshwater lakes is a global problem and has been a matter of concern to the
public for several decades. In lakes where the external P loading has been reduced, internal P loading (sediment-
P release to surface waters) is widely recognized as the key factor affecting lake trophic status and delaying lake
recovery from eutrophication (Burley et al., 2001; Trolle et al., 2010). Considerable work has been done on
sediment-P speciation to evaluate sediment-P release potentials and implications for lake restoration management
(Gonsiorczyk et al., 1998; Ribeiro et al., 2008).
The paradigm that oxygen controls the sediment-P release via reductive dissolution of Fe-P fraction in surface
sediments has been accepted as the classical model for a long time (Einsele, 1936, 1938; Moosmann et al., 2006),
which was supported by numerous short-term (days or seasonal) laboratory or in-situ studies (Chen et al., 2018).
Based on this paradigm, it was assumed that an oxic sediment–water interface might limit the release of Fe-P from
sediments and, therefore, improve P retention in lake sediments. However, the restoration measures with artificial
hypolimnetic oxygenation/aeration applied in eutrophic lakes proved to have only short-lasting effects but no direct
effects on internal P loading and redox-dependent sediment-P retention on longer terms (Gächter, 1987; Gächter
and Wehrli, 1998; Moosmann et al., 2006; Hupfer and Lewandowski, 2008). Gächter and Müller (2003) and
Moosmann et al. (2006) further argued that, on multi-decadal or longer time scales, P retention in lake sediments
might eventually primarily depend on the P-binding capacity of anoxic sediments and sediment composition (e.g.
Fe, Mn, Al, and Ca contents). Nevertheless, until now, there is a lack of evidence from well-dated sediment cores,
and there is still a need to know which processes may have a dominant influence on sediment P-fraction retention
on longer time scales (e.g., decades or more). This information is crucial for predicting and, ultimately, managing
sediment-P release, especially in deep lakes, because hypolimnetic anoxia in deep lakes can lead to large loads of
sediment-P release. Furthermore, eutrophication has been demonstrated to affect sediment-P release via controlling
hypolimnetic anoxia and lake mixing regime in seasonally stratified deep lakes (Tu et al., 2019). It is not yet fully
understood whether and how lake trophic levels and hypolimnetic anoxia can influence the long-term behavior of
sedimentary P-fraction retention in deep lakes.
The restoration technique of hypolimnetic water withdrawal has been frequently applied in seasonally stratified
lakes in Europe (Kucklentz and Hamm, 1988; Nürnberg, 2007), whereby P-enriched water from the hypolimnion
is discharged directly into the lake outflow. This restoration technique has been shown to efficiently reduce P
concentrations in lake waters (Nürnberg, 2007). However, the long-term influence of this restoration on
sedimentary P-fraction retention is unclear.
The objectives of this study were to (1) explore the main factors controlling long-term changes of P-fraction
retention in sediments of deep lakes, (2) investigate how sediment P-fraction retention responds to changes in lake
eutrophication and hypolimnetic anoxia of the past prior to anthropogenic eutrophication, and (3) examine the
effects of lake hypolimnetic withdrawal restoration on sedimentary P-fraction retention. To achieve these
objectives, we investigated short sediment cores from Lake Burgäschi, a deep and eutrophic lake on the Swiss
Plateau. Sedimentary green-pigments s (chlorophylls and diagenetic products) inferred from hyperspectral imaging
(HSI) scanning and XRF-inferred Fe/Mn ratios primarily reflect lake trophic state evolution (aquatic primary
productivity) and hypolimnetic oxygenation, respectively. A sequential P-extraction with five P fractions was



performed to uncover P fractionation in sediment profiles. We combined all data to identify the dominant factors
responsible for temporal changes in P-fraction retention. Changes in P-fraction records for the periods before and
during the restoration were also investigated.
Lake Burgäschi is an excellent study site because there were substantial changes in lake trophic levels and possibly
lake-mixing regimes since the last century (Guthruf et al., 1999; van Raden, 2012), and exceptionally long
historical and limnological survey data are available for most of the last 50 years. Hypolimnetic withdrawal
restoration has operated in the lake since 1977.

## 83  2 Study site

Lake Burgäschi (47°10′8.5″N, 7°40′5.9″E) is a small lake located on the Swiss Plateau (Fig. 1a). It has a very
restricted catchment (3.2 km$^2$). The catchment area geologically belongs to the Molasse Basin, and mostly consists
of carbonate-rich sandstones and mudstones (Schmid et al., 2004). The kettle hole lake was formed after the retreat
of the Rhone glacier (ca. 19 k yr. BP; Rey et al., 2017) and, today, has a maximum water depth of ~31 m, which
is quite deep in contrast to the small surface area of 0.21 km$^2$ (Guthruf et al., 1999). The mean retention time of
the lake water is ~1.4 year (Nürnberg, 1987). The lake has several small inflows in the southwest (Rey et al., 2017)
and one outflow in the north (Fig. 1c).
Since the 19$^{th}$ century, the lake's water level was lowered several times to create agricultural lands, with the most
lowering (up to 2 m) during 1943-1945 (Guthruf et al., 1999). Agricultural area currently covers ~55% of the lake
catchment, followed by ~29% area of forests. The lake region experiences a warm humid continental climate (Dfb;
Köppen-Geiger classification). The mean annual temperature is 9 °C and the warmest month is July (mean
temperature 19 °C).
Lake Burgäschi has been highly productive (eutrophic to highly eutrophic state) since the 1970s with high algal-
biomass production and anoxic conditions in the hypolimnion (Guthruf et al., 1999, 2013). The eutrophication in
Lake Burgäschi has been linked to increased agricultural P inputs via drainage into the lake in the second half of
the 20$^{th}$ century (Guthruf et al., 1999). To mitigate the eutrophication, hypolimnetic withdrawal restoration has
been applied in Lake Burgäschi since 1977, and the lake water has been monitored twice a year for more than 30
years for various parameters, such as pH, oxygen content, phosphorus concentrations, phytoplankton biomass etc.
Despite a sharp decline in hypolimnetic phosphorus concentrations due to the restoration, a high production of
algae biomass continues today. Additionally, hypolimnetic oxygenation conditions and the lake trophic state have
been stabilized but not fundamentally improved (Markus, 1995; Guthruf et al., 2013).

## 106  3 Materials and methods

### 107  3.1 Core collection and sampling

In September 2017, two 75-cm-long sediment cores (Burg17-B and Burg17-C) were retrieved from the deepest
point of Lake Burgäschi (water depth ~31 m) (47°10'8.6"N, 07°40'5.3"E; coring site in Fig. 1c) using a UWITEC



gravity corer. After the collection, the cores were stored in a dark cold room (~4 °C). After opening and splitting lengthwise, core-half A of Burg17-B was continuously subsampled at 2-cm resolution from 0 to 60 cm for $^{210}$Pb and $^{137}$Cs dating (section 3.2). The oxidized surface of core-half B (Burg17-B) was visually described (Schnurrenberger et al., 2003) before non-destructive XRF core and HSI scanning (Section 3.3). After the opening, one-half of core Burg17-C was transferred immediately into a glove box with an anoxic atmosphere where it was continuously subsampled at 2-cm resolution from 0 to 72 cm. The fresh sediments from each sample slice were homogenized and used for sequential P extraction. Afterwards, the remaining sediment was freeze-dried and homogenized for bulk element analyses (Section 3.4).

**3.2 Chronology**

The chronology of the core Burg17-B is based on $^{210}$Pb and $^{137}$Cs activity profiles. The freeze-dried and homogeneous samples were stored dry and dark until analysis. The $^{210}$Pb, $^{137}$Cs and $^{226}$Ra radiometric activities were measured by gamma spectrometry conducted at the Department of Chemistry and Biochemistry at University of Bern. 1.3-5.1 g of the freeze-dried samples were encapsulated into polystyrene petri dishes (68 mm O.D., 11 mm height; Semadeni, Ostermundigen, Switzerland) together with a polystyrene disk to fill in the headspace above the material and the petri dishes were vacuum-sealed into a gas-tight aluminum foil. $^{210}$Pb (46.5 keV), $^{241}$Am (59.5 keV), $^{226}$Ra progenies $^{214}$Pb and $^{214}$Bi (295.2, 351.9 and 609.3 keV), as well as $^{137}$Cs (661.7 keV) were measured using a Broad Energy Germanium (BEGe) detector (Canberra GmbH, Rüsselsheim, Germany). This system is composed of a high-purity germanium crystal of 50 cm$^2$ area and 30 mm thickness with a 0.6 mm thick carbon epoxy window, which shows high absolute full-energy peak efficiencies for close on-top geometries of >20% and ~5% for $^{210}$Pb and $^{137}$Cs, respectively. Low integrated background count rates of 0.20 s$^{-1}$ (energy range of 30-1800 keV) were achieved by application of low-background materials, installation in third underground floor (~10 m of water-equivalent overburden), passive shielding (outside to inside: 10 cm low-background lead, 3 mm ancient lead with negligible $^{210}$Pb content, 2 mm cadmium), flushing of the shield interior with nitrogen gas and an active anti-cosmic shield (plastic scintillator panels of totally 1 m$^2$ area mounted directly above the passive shielding). Supported $^{210}$Pb in each sample was assumed to be in equilibrium with the in-situ $^{226}$Ra (equilibration time 4 weeks). Unsupported $^{210}$Pb activity was calculated by subtracting $^{226}$Ra activity from total $^{210}$Pb activity level-by-level. The correction for the total unsupported $^{210}$Pb missing inventory followed Tylmann et al. (2016).

The $^{210}$Pb chronology of Core Burg17-B was determined using the Constant Rate of Supply (CRS) model (Appleby, 2002), which accounts for variation in sediment accumulation rates. We tested two CRS models: CRS-1 model was unconstrained (i.e. without reference points from the $^{137}$Cs activity). The CRS-2 model was constrained with the chronologic marker of peak fallout from nuclear weapons testing in 1963 ($^{137}$Cs and $^{241}$Am). Both models were then tested and validated with independent time-markers at the onset of nuclear weapons testing in 1953/54 and the Chernobyl accident in 1986/87 (onset of $^{137}$Cs and peak of $^{137}$Cs and $^{241}$Am, respectively).

The two sediment cores (Burg 17-B and Burg 17-C) are visually very similar but show a length-offset due to coring compaction of approximately 2-6 cm (Fig. S1). The age-depth stratigraphy of Burg17-C core was inferred from the dated core Burg17-B by visual stratigraphic correlation from high-resolution core pictures.



### 3.3 Non-destructive geochemical methods

Non-destructive X-ray fluorescence (XRF) core scanning was done using an Avaatech XRF Core Scanner (Richter et al., 2006) for semi-quantitative element composition measurements at 0.5 mm resolution to capture relative elemental concentrations of the laminae. Core surface was smoothed and covered with a 4-μm-thick Ultralene foil prior to the analysis. Elements were measured using a Rhodium anode and a 25 μm Be window. The lighter elements were measured at 15 seconds count time at 10 kV with 1500 A, no filter; while the heavier elements were exposed at 40 seconds at 30 kV with 2000 A, Pd-thin filter. Element intensities (semi-quantitative concentrations) of the selected elements (Mg, Si, Al, K, Ti, Rb, P, Fe, Mn, Ca) are expressed as count rates (counts per second, cps).

Following the methodology in Butz et al. (2015), hyperspectral imaging (HSI) scanning was performed using a Specim Ltd. Single Core Scanner equipped with a visual to near infrared range (VNIR, 400–1000 nm) hyperspectral linescan camera (Specim PFD-CL-65-V10E). Parameters were set for a spatial resolution of ~70 μm/pixel and a spectral sampling of 1.57 nm (binning of 2). Spectral endmembers were determined using the "Spectral Hourglass Wizard" of the ENVI 5.5 software package (Exelisvis ENVI, Boulder, Colorado). The relative absorption band depth (RABD) index calculation was performed following the method in Schneider et al. (2018). However, based on the spectral end members (Fig. S2), we used the absorption feature between the wavelengths R590 and R765 (590-765 nm), i.e. RABD$_{590-765}$. Butz et al. (2017) and Schneider et al. (2018) revealed that this index is well calibrated to absolute green-pigments concentrations (chlorophyll $a$ + pheophytin a) in sediments. Therefore, in our study, the relative concentrations of green-pigments inferred from RABD$_{590-765}$ index values provide a semi-quantitative reconstruction of lake primary productivity (total algal abundance) at sub-annual resolution, and are suggested to reflect the trophic state evolution of Lake Burgäschi.

### 3.4 Phosphorus fractionation scheme and bulk elements analyses

The P-fractionation extraction protocol principally follows the four-step extraction protocol in Tu et al., (2019). In addition, we added the last extraction step from Lukkari et al. (2007) to determine refractory organic P (F5). This P fraction (F5) is practically biologically unavailable and subject to permanent P burial. The first four fractions are NaCl-TP (F1: loosely bound P), NaBD-TP (F2: redox-sensitive Fe- and Mn-bound P), NaOH-TP (F3: Al- and Fe-bound P), and HCl-TP (F4: Ca-bound P) (Tu et al., 2019), whereby NaCl-TP, NaBD-TP and NaOH-TP fractions together as considered relatively labile P fractions because they may release P back to the water column under anoxic or high pH environments (Rydin, 2000). The HCl-TP and refractory organic P (Ref.-P$_o$) fractions are classified as relatively stable or refractory P fractions. Total P in sediments was obtained from the sum of the five P fractions. The P in extract samples was measured by inductively coupled plasma mass spectroscopy (7700× ICP-MS) (Agilent Technologies, Germany) after the dilution with nitric acid (HNO$_3$) to reach a final concentration of 1% v/v HNO$_3$.

Concentrations of total carbon (TC), total nitrogen (TN), and total sulfur (S) in sediment samples were determined using an Elementar vario El Cube elemental analyzer. Total inorganic carbon (TIC) content was calculated by multiplying loss on ignition at 950 °C (LOI$_{950}$, following the method proposed by Heiri et al. (2001)) by 0.273, i.e. the ratio of the molecular weight of C and CO$_2$. Total organic carbon (TOC) content was calculated using the



equation TOC =TC-TIC. Sediment dry bulk density and water content were determined following the method in
Håkanson and Jansson (2002).
**3.5 Data analyses**
Multivariate statistical analyses were performed with R version 3.4.2 (R Development Core Team, 2017). Prior to
data analyses, RABD$_{590-765}$ index values (resolution 70 μm) were aggregated to a spatial resolution of 0.5 mm (the
spatial resolution of XRF data). Stratigraphically constrained incremental sum of squares clustering (CONISS;
Grimm, 1987) was then performed on semi-quantitative proxies (i.e. RADB$_{590-765}$ index and XRF-element data)
with R-package ''rioja'' (Juggins, 2017). The number of significant clusters was determined with a broken-stick
test (Bennett, 1996). A principal components analysis (PCA) was performed on the centered and standardized data
of semi-quantitative proxies, using the "Vegan" package (Oksanen et al., 2013). Then after, XRF-element and
RABD$_{590-765}$ index data points within the depth range corresponding to samples taken from core Burg17-C were
used to calculate mean values for each sample. In order to identify the primary factors influencing the variations
in sedimentary P fractions, a redundancy analysis (RDA) was performed on the centered and standardized dataset
of P fractions (response variables) and other sediment geochemical parameters (explanatory variables) with the
"vegan" package. In the RDA computation, the correlation matrix option was selected and the scaling was
conducted on a correlation biplot.
**4 Results**
**4.1 $^{137}$Cs and $^{210}$Pb chronology**
The two distinctive peaks of $^{137}$Cs in sediment profiles are detected at 31 cm and 15 cm depths (Fig. 2b),
corresponding to the 1963 and 1986 major fallout events, respectively (Appleby, 2002). Furthermore, $^{241}$Am
activity peaks at the same depths (Fig. 2b), confirming that the 1963 and 1986 $^{137}$Cs peaks were due to atmospheric
fallouts (Michel et al., 2001). The first traces of $^{137}$Cs occur at 37 cm depth, indicating the first widely detectable
fallout from atmospheric nuclear testing in 1953/1954 (Pennington et al., 1973).
The $^{210}$Pb activity in Core Burg17-B shows a relatively monotonic decrease down to a sediment depth of 17 cm.
Further down, larger variations are found (Fig. 2a). The $^{210}$Pb and $^{226}$Ra activities do not reach the equilibrium;
unsupported $^{210}$Pb activity in the oldest sample (59 cm) is still above the limit of detection (14.0±6.8 Bq·kg$^{-1}$). The
observed cumulative inventory of unsupported $^{210}$Pb is 2941 Bq·m$^{-2}$. We corrected this value (missing inventory
correction; Tylmann et al., 2016) by applying an exponential equation using the lowermost values of cumulative
dry mass and unsupported $^{210}$Pb activity between 8 and 60 cm depths. As a result, a correction value of 125.2
Bq·m$^{-2}$ (missing inventory) is added to the final total unsupported $^{210}$Pb inventory (3066 Bq·m$^{-2}$).
The CRS-2 model (constrained through 1963) shows a better agreement with the independent $^{137}$Cs markers at
1953/54 and 1986/87 than the CRS-1 model (Fig. 2c). Therefore, CRS-2 model results were chosen for determining
the age-depth profile and sediment mass accumulation rates (MAR) of Core Burg17-B. The mean age at 59 cm
sediment depth dates back to ~1930. The extrapolated mean age at 61 cm depth is ~1926 calculated using the mean
sediment accumulation rate between 54-60 cm (2 yr·cm$^{-1}$).



### 4.2 Sediment lithology, green-pigments (RABD$_{590-765}$ index) and XRF-element records

Four sediment facies (I to IV, Fig. 3a and 4) are identified based on visual classification and the CONISS-analysis results of XRF-element intensities.

In Zone I (75.4-61cm, pre ~1926), the sediments consist of visible thin brown-to-reddish laminae (Mn- and Fe rich). Green-pigments concentrations inferred from RABD$_{590-765}$ index values show a homogenous distribution with the lowest values within the sediment profile (Fig. 2d). Fe/Mn ratios vary within very low values (mostly below 10). The Mn, Fe, P and Fe/Ti values show high levels with large variability. Extremely low Ca amounts are noted in this zone.

In Zone II (61-34cm, ~1926-1960), the sediments show dark gyttja, partly laminated with light Ca-rich layers. Green-pigments concentrations slightly increase yet still show little variability. A sharp increase of green-pigments concentrations occurs at 60 cm, and the first two local peaks near 55 cm (~1938) and 48 cm (1945) are notable. Fe/Mn ratios remain at slightly higher values than in Zone I. The Mn, Fe, P contents and Fe/Ti values all decline to low levels and remain relatively stable. Ca counts increase gradually over the whole Zone II.

In Zone III (34-21.5 cm, ~1960-1977), the sediments are mostly characterized by brown-to-reddish laminations (Mn-Fe rich), with thicker and more distinct laminae contacts than in Zone I. Green-pigments concentrations exhibit much higher values with positive trends, intensified variability, and several maxima (seasonal algal blooms). Fe/Mn ratios first drop in the lower part (34-27 cm) and then continue to increase upward to the top-part of Zone III. Fe, Mn, P, and Fe/Ti values change with general opposite trends to Fe/Mn ratios. Ca contents are elevated during this period relative to Zones I and II.

In Zone IV (21.5-0 cm, ~1977-2017), the sediments exhibit a clear laminated structure with much more pronounced light calcite layers. The laminations are characterized by a regular succession of light calcite layers (Ca-rich) and dark organic-rich layers (Fig. S3). Green-pigments concentrations display the highest levels with large fluctuations, and reaches distinct local maxima at 18 cm (1981), 15 cm (1985), 13 cm (1987), 12 cm (1988), and 8 cm (1997) depths (Fig. 3b). Fe/Mn ratios remain at similarly high values as in Zone II, yet with more variability. The Fe, Mn, and P element counts and Fe/Ti all show constantly very low values. The Ca amounts are the highest in the profile and show considerabley variability.

Two principle components, PC1 and PC2 were shown to be significant using a broken stick model. They explain ~35 % and ~30 % of the total variance in the dataset, respectively (PCA-biplot; Fig. S4). The PC1 has strong positive loadings for the terrigenous elements (K, Ti, Rb etc.) and thus represents mainly erosional processes related to allochthonous inputs. The PC2 has strong positive loadings for redox-sensitive elements (Fe, Mn), P and Fe/Ti, but negative loadings for Ca, Fe/Mn ratios and green-pigments index values. Therefore, PC2 reflects changes in redox conditions of hypolimnetic water and lake primary productivity. The results of additional PCA analyses zone by zone (Fig. S5b) show that Mn, Fe and P were mostly independent of terrigenous elements (in Zones I to III), however in Zone IV, Mn, Fe and P become correlated with the terrigenous elements. The vertical profile of XRF-P matches very well with the changes of total P concentrations in sediments (Fig. S6). It reveals





that XRF-P data can reliably represent qualitative variations of total P concentrations in sediment profiles of Lake
Burgäschi.

### 4.3 Bulk elements and P fractions in sediment profiles

Sediment TIC, TOC, TOC/TN ratio, S and P fractions also show distinctive features along the four stratigraphic
zones (Fig. 5). From the upper part of Zone I (65.2-61 cm; ~1926) to Zone IV, TIC shows a similar pattern to the
XRF-Ca contents (Fig. 4 and 5) suggesting that TIC is mostly present in the form of $CaCO_3$. Over the whole profile,
TOC/TN ratios are within the range of 9-11. TOC and TOC/TN ratios exhibit mostly similar patterns from Zone I
to Zone III. By contrast, total sulfur (S) contents display a different pattern, showing very low values in Zone I
and II (mean ~0.5%), and a substantial increase in Zone III and IV.

The concentrations of relatively labile P fractions (i.e. NaCl-TP, NaBD-TP and NaOH-TP) and total P have a
similar trend over the whole profile (Fig. 5 and 6a). They all display rather large values during the upper part of
Zone I and generally reduced values in Zone II. Afterwards, they increase to peaks at ~25 cm depth but sharply
decrease to the lowest values in the upper boundary of Zone III and throughout Zone IV. HCl-TP and Res.-$P_o$
fractions vary differently compared with the other fractions. Low contents of HCl-TP fraction are observed in
Zone I and II. HCl-TP fraction has a rather similar pattern as labile P fractions in Zone III, but then it remains at
high levels in Zone IV. Res.-$P_o$ fraction contents show relatively stable values from Zone I to Zone II, followed
by a gradual rise in Zone III and in the upper part of Zone IV.

Regarding the P composition in sediment profiles (Fig. 6), from Zone I to Zone III (65.2-21.5 cm) NaBD-TP
fraction is the most important P-form representing ~50% of total P followed by NaOH-TP fraction. However, in
Zone IV (depth above ~ 21.5 cm), HCl-TP becomes the main P fraction (39% of total P) over NaBD-TP (30% of
total P).

The relationships between response variables and explanatory variables are visible on the redundancy analysis
(RDA) biplot (Fig. 7), which, in most cases, correspond well to the results of Spearman rank correlation test (Fig.
S7). The relatively labile P fractions (NaCl-TP, NaBD-TP and NaOH-TP) and total P in sediments are strongly
positively correlated with redox-sensitive elements (Fe and Mn) and autochthonous Fe (Fe/Ti). However, these P
fractions are negatively related to hypolimnetic oxygenation proxy (Fe/Mn ratios) and, to some extent, to lake
productivity indicators (green-pigments, XRF-Ca and TIC). HCl-TP and Ref.-$P_o$ fractions are positively correlated
with each other. However, only HCl-TP fraction has close positive relationships with lake productivity indicators.

### 5 Discussion

### 5.1 Trophic state evolution of Lake Burgäschi

Four main phases of different lake trophic levels (based on $RABD_{590-765}$ index record) were distinguished since the
early 1900s. During the period prior to ~1926 in Zone I, the lowest green-pigments index values reflect low lake
primary productivity. In the early 1900s, agricultural impacts around the catchment area of Lake Burgäschi were



not prominent (Guthruf et al., 1999). It can be expected that the lake had low nutrient loads from the catchment
drainage during this period. Lake Burgäschi is classified as naturally oligotrophic based on morphometric
parameters (LAWA, 1998) to naturally mesotrophic according to Binderheim-Bankay (1998). Therefore, at the
times of Zone I, Lake Burgäschi was likely in low trophic levels with a possible oligotrophic-mesotrophic
condition.
The transition to Zone II (~1926-1960) was marked by generally increased sedimentary green-pigments and
CaCO$_3$ contents (Fig. 4 and 5), indicating enhanced lake primary productivity. The slightly decreased TOC/TN
ratio also suggests a likely rise in autochthonous organic matter proportion (Meyers and Ishiwatari, 1993). The
first two algal blooms (peaks of green-pigments index; Fig. 3d) imply a very likely mesotrophic to eutrophic state
of the lake. Indeed, the study of Büren (1949) revealed that in 1943-1945, the trophic state of Lake Burgäschi had
already shifted between mesotrophic and eutrophic.
In Zone III (~1960-1977), continuously increasing green-pigments concentrations and several algal bloom events
reveal strong positive trends in lake eutrophic levels. The significant eutrophication in Lake Burgäschi might have
caused intensified CaCO$_3$ precipitation and sulfur (S) deposition to sediments (Fig. 4 and 5), which is in agreement
with the findings from many other eutrophic lakes (Bonk et al., 2016; Holmer and Storkholm, 2001; Schneider et
al., 2018).
During Zone IV (1977-2017), we interpret that Lake Burgäschi was in highly eutrophic conditions, based on
constantly high green-pigments index values and multiple prominent algal blooms (Fig. 4). Low and decreasing
TOC/TN ratio values (< 10) in this zone suggest a dominant source of organic matter in sediments from aquatic
primary production, which has been interpreted as a signal of eutrophic waters (Enters et al., 2006). Our
interpretation is further supported by high chlorophyll-*a* concentrations in surface waters (>8 ug L$^{-1}$; Markus, 2007)
and the dominance of blue-green algae in the phytoplankton biomass during 1977 to 1992, which characterized
Lake Burgäschi as highly eutrophic (Guthruf et al., 2013; Markus, 1995).

### 5.2 Reconstruction of hypolimnetic oxygenation regimes of Lake Burgäschi

A large number of studies have used the proxy of Fe/Mn ratios in sediments to reconstruct past water oxygenation
and mixing regimes of the lake, such as Frugone-Álvarez et al. (2017), Mackereth (1966), and Żarczyński et al.
(2019) etc. However, this proxy and its interpretation are limited to cases in which the annual cycle of Fe and Mn
deposition in lakes is mostly driven by redox changes in the hypolimnion and related diagenetic processes in
surface sediments instead of driven by terrestrial inputs (Boyle, 2001; Naeher et al., 2013). In Lake Burgäschi,
during Zone I to III, Mn and Fe were mostly independent of erosion indicators as shown in Fig. S5b. Furthermore,
Van Raden (2012) has revealed that the presence of Mn-rich laminae in sediments of Lake Burgäschi can indicate
frequent short-term wind-induced mixing events in the lake. Therefore, we suggest that the deposition of Fe and
Mn during these three zones was mainly controlled by in-lake processes. Fe/Mn ratios together with Mn
precipitation is reliably tracking past changes of hypolimnetic oxygenation of Lake Burgäschi.
In Zone I (pre ~1926), the sediments feature well-preserved Mn-Fe rich laminations and very low Fe/Mn ratios
(Fig. 3 and 4), suggesting that the lake hypolimnion was seasonally well-oxygenated. The similar occurrence of



visible Mn-and Fe rich laminae in sediments were also reported by Rey et al. (2017) and Van Raden (2012) in
Lake Burgäschi and from other lakes, for example, Lake of the Clouds in the US (Anthony, 1977), Lake Cadagno
in the Swiss Alps (Wirth et al., 2013), and Lake Żabińskie in Poland (Żarczyński et al., 2018). They revealed that
the red-orange Mn-rich layers mostly consist of authigenic rhodochrosite ($MnCO_3$) that was formed when Mn-rich
anoxic bottom waters are mixed with oxygenated surface waters for short intervals. The preservation of this Mn-
rich layer is only possible when its sedimentation process exceeds the release process under anoxic hypolimnetic
conditions (Stevens et al., 2000). Therefore, during this period, short-term mixing events and associated
oxygenation may have occured during overall stratified or anoxic conditions in the hypolimnion.

In Zone II (~1926-1960), the higher Fe/Mn ratios and very low Mn- and autochthonous Fe (Fe/Ti) amounts are
interpreted as the results of stable anoxic hypolimnetic waters. The formation and preservation of Fe-and Mn-
oxides in sediments can be largely prevented under long-term stratification/reducing conditions (Stevens et al.,
2000). The lake most likely developed anoxic hypolimnetic conditions with yearly incomplete or missing
circulation in the hypolimnion.

In Zone III (1960-1977), overall decreased Fe/Mn ratios combined with reappearing Mn-and Fe-rich laminations
reflects better short-term oxic conditions in hypolimnetic waters than in Zone II. However, during ~1970 to 1977,
Fe/Mn ratios gradually increased (Fig. 4), which points to less oxic conditions in the hypolimnion. It seems to be
related to synchronously progressive lake eutrophication. Higher primary productivity and strengthened anoxia in
the hypolimnion are commonly observed in stratified lakes (Giguet-Covex et al., 2010; Mikomägi et al., 2016).
Higher lake primary productivity increases high-rate aerobic degradation of organic matter and, consequently,
oxygen-depletion in the hypolimnion and sediments (Gächter and Müller, 2003; Nürnberg, 2007).

Finally, in Zone IV (1977- 2017) Fe/Mn ratios proxy is no longer valid to indicate hypolimnetic oxygenation
regime, as suggested by predominatly terrestrial sources of sediment Fe and Mn (Fig. S5b). Nevertheless, the well-
preserved laminated sediments during this period are a clear sign of no benthic bioturbation and thus represent an
indicator of generally strong anoxic conditions in hypolimnetic waters, occurring simultaneously with a highly
eutrophic period. According to the limnological monitoring data of Lake Burgäschi between 1978 and 2007
(Markus, 2007), the lake water was completely anoxic at depths below 20 m during the summer-autumn
stratification; even during winter circulation of most years, the lake water was still not completely mixed.

**5.3 Phosphorus composition and factors controlling long-term P-fraction retention in sediments**

Prior to 1977 (i.e. Zones I-III), NaBD-TP (redox-sensitive Fe- and Mn bound P) and NaOH-TP (partly non-
reducible Fe oxides-P) fractions were the primary P forms in sediments of Lake Burgäschi. This seems to compare
well with the study of Moosmann et al. (2006), who suggested that sediment Fe contents controls P retention in
sediments of the Swiss Plateau lakes. However, after ~1977, we observed a change to predominantly Ca-P (apatite-
P), occurring concurrently with considerable $CaCO_3$ precipitation during this highly eutrophic period (Fig. 4; Sect.
5.1). We interpret this as an incidence of biologically driven co-precipitation of Ca and P in highly productive





lakes. The phenomenon of Ca-P co-precipitation has been observed and studied in many calcareous lakes (Dittrich
and Koschel 2002; Whitehouse, 2010), and is assumed to be responsible for the scavenging of dissolved P from
surface waters of eutrophic lakes (Hamilton et al., 2009). Furthermore, large amounts of Ca-P in surface sediments
(top 21 cm) can act as a potential negative feedback to eutrophication in Lake Burgäschi. This is because Ca-P is
relatively stable in sediments and has low potentials of P-release from surface sediments back to lake waters.

In the sediment profile of this study (coving more than 100 years), retention of total P and labile P fractions was
mainly controlled by autochthonous Fe (Fe/Ti), Mn, and hypolimnetic oxygenation proxy (Fig. 7). Our results
support the previous suggestion that long-term permanent sediment-P retention is largely limited by the sediment's
binding capacity in anoxic conditions (Hupfer and Lewandowski, 2008; Moosmann et al., 2006), which,
specifically in our case, is determined by redox-sensitive elements (autochthonous Fe and Mn) preserved in
sediments. These findings are discussed in the context of each cluster zone as follows. During Zone I and Zone III,
when the hypolimnion had better oxic conditions (see Sect. 5.2), the increased retention of Mn and Fe, and labile
P fractions occurred simultaneously (Fig. 4 and 5). This phenomenon might be caused by efficient P-trapping in
Mn- and Fe enriched layers. It has been suggested that the formation of laminated Mn- and Fe enriched layers
could serve as a protective cap to reduce P release from surface sediment layers to the anoxic hypolimnion
(Żarczyński et al., 2018), which thus can help improve P retention within these sedimentary layers. In Zone II,
small amounts of labile P fractions might result from reduced P-bearing solid phases (Mn and Fe oxyhydroxides)
in sediments under more anoxic conditions in the hypolimnion (higher Fe/Mn ratios; Sect. 5.2). In Zone IV,
however, the lowest retention of total P and labile P fractions in recent sediments was noteworthy (Fig. 5 and 6).
We interpret this as a combined result of eutrophication-induced hypolimnetic anoxia and lake restoration by
discharge of P-rich hypolimnetic water since 1977. On the one hand, under stable anoxic conditions in the
hypolimnion caused by strong eutrophication, reduced Mn and Fe preservation (Fig. 4) suggests a low capacity of
permanent P-trapping within the anoxic sediments. On the other hand, hypolimnetic withdrawal restoration in
Lake Burgäschi has substantially reduced hypolimnetic P concentrations by a factor of 5-6 since 1978 (Markus,
2007). This indicates a concomitant decrease in sediment-P release to the hypolimnion and P sedimentation to
water-sediment interface as well. Consequently, decreased P retention in sediments was observed.

In the whole sediment profile, HCl-P and Ref.-$P_o$ fractions had mainly autochthonous origins (Fig. 7). HCl-P
fraction retention, to a large extent, resulted from authigenic $CaCO_3$-P precipitation, and increased with higher
eutrophic levels in Zone III and IV (Fig. 5; Sect. 5.1). Interestingly, HCl-P fraction retention in sediments was
generally lower in Zone IV (Fig. 5) than in Zone III, although the lake in Zone IV had relatively higher eutrophic
levels. The pH in the hypolimnion of Lake Burgäschi varied between 7.0 and 7.5 according to the monitoring data
in 1993, 2003, and 2013 (Guthruf et al., 2013). Therefore, the acid dissolution of Ca-P in hypolimnion and at the
water-sediment interface is small and unlikely significant during Zone IV. The generally decreased Ca-P retention
was seemingly related to the lower water-P concentrations due to hypolimnetic withdrawal as discussed above,
thus resulting in smaller amounts of $CaCO_3$-P co-precipitation in the epilimnion. Overall, Ref.-$P_o$ fraction retention
in the sedimentary profile shows less variability compared with other P fractions (Fig. 5). Nevertheless, in the
upper sediments (top 12 cm; Fig. 5) the enhanced retention of Ref.-$P_o$ fraction could be derived from increased
algal refractory organic matter.




The interesting fact is that the water-P reductions caused by the hypolimnetic withdrawal in Lake Burgäschi have
been ineffective in reducing algal blooms and curbing eutrophication. Similar findings were also reported from
some lakes in Europe and the US (Kosten et al., 2012; Kolzau et al., 2014; Fastner et al., 2016). These authors
have attributed this phenomenon to insufficient P-load reduction, higher water temperatures under global warming
of the last few decades, and the light or nitrogen limitation of surface-water phytoplankton. We suggest that these
factors mentioned may also play a role in promoting currently high primary productivity in Lake Burgäschi, but
the main driver is, to our knowledge, unclear.

**6 Conclusion**
The primary productivity of Lake Burgäschi started to increase from the 1930s and the eutrophication proceeded
since the 1960s. The hypolimnion oxygenation regime of Lake Burgäschi was characterized by four major phases
since the early 1900s. Persistent anoxic conditions in the hypolimnion after ~1977 coincide with highly eutrophic
conditions.

The dominant factor controlling the long-term retention of labile P-fractions and total P in sediments of Lake
Burgäschi was found to be autochthonous Fe and Mn contents in anoxic sediments, which were controlled by
redox conditions in the hypolimnion. Importantly, we found substantially decreased retention of labile P fractions
and total P after ~1977. We attributed this phenomenon to a combined result of eutrophication-induced anoxic
conditions in the hypolimnion and hypolimnetic withdrawal restoration. The former reduced the P-binding
capacity (Mn and Fe oxyhydroxides) of sediments and the latter has decreased hypolimnetic P-concentrations
(external P loading). By contrast, refractory Ca-P fraction predominated in surface sediments after ~1977, which
indicates a decreased sediment-P availability and potential of sediment-P release (internal P loading). These results
together imply that although hypolimnetic withdrawal can effectively reduce external and internal P loadings in
Lake Burgäschi, it still has no significant impact on decreasing eutrophication. We attribute the delaying of lake
recovery to various factors such as still high nutrient inputs from the nearby or surrounding agricultural area, light
and/or nitrogen limitation to lake phytoplankton, and warmer lake temperature due to global warming. This study
calls for consistently more effective measures to minimize external P loadings from the catchment, such as
optimizing fertilizer application practices and technical measures in the settlement drainages.



**Data availability**
The data will be made available at on PANGAEA after the manuscript is published.


**Author contributions**
L.T. helped with sample collection, analyzed the sediment, conducted data analysis, wrote the manuscript, and
acquired the funding for the project.





P. Z. helped with sediment core subsampling, conducted the hyperspectral imaging (HSI) scanning and XRF-
scanning, substantially contributed to the data interpretation.
S.S. measured gamma-spectroscopy radiometric activities, generated the data for chronology and helped with
data interpretation.
R.L. conducted the XRF-scanning and helped with data interpretation.
M.G. designed the study, helped discussing the results and supervised the project.
All authors commented on the manuscript and approved the final version of the manuscript.

**Competing interests**
The authors declare that they have no conflict of interest.

**Acknowledgements**
We thank Stamatina Makri and Dr. Andre F. Lotter for their help during the fieldwork. We thank Irene Brunner,
Patrick Neuhaus, Dr. Daniela Fischer and Andrea Sanchini for their expertise and the lab assistance. Further, we
acknowledge Dr. Klaus A. Jarosch for the valuable suggestions about phosphorus data. The project was funded
by the Swiss National Science Foundation under the grant number 200021-172586, a Fellowship Grant from the
Chinese Scholarship Counsel and the International PhD Fellowship from University of Bern.

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



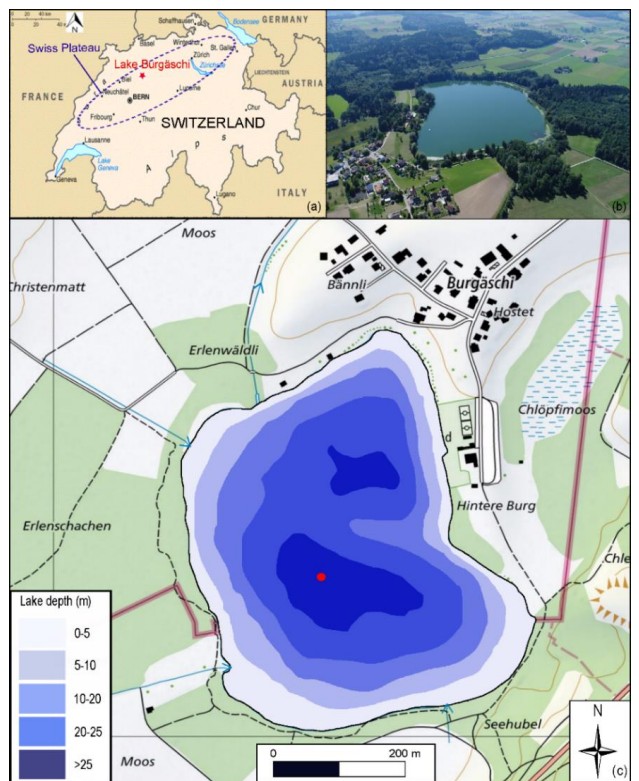

**Figure 1: Study site. (a) Overview map of Switzerland and the Swiss Plateau. Lake Burgäschi is indicated as the red**
**asterisk. (b) Satellite photo of Lake Burgäschi and catchment (© 2018 Google Maps). (c) Bathymetric map of Lake**
**Burgäschi adapted from Guthruf et al. (1999). The red dot indicates the coring site (color figure online). Green areas**
**around the lake indicate forests, white areas agricultural lands. Inflow and outflow to the lake are indicated by blue**
**arrow lines (topographic maps: © swisstopo).**
















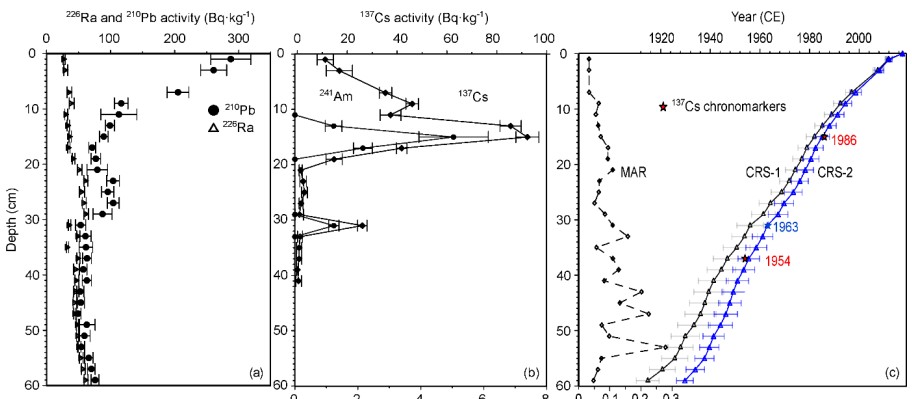


**Figure 2: (a) Total $^{210}$Pb, $^{226}$Ra, and (b) $^{137}$Cs and $^{241}$Am activity concentration profiles in sediment core Burg17-B from Lake Burgäschi; (c) The comparison of different $^{210}$Pb CRS models: unconstrained CRS -1 model and constrained CRS-2 model at 1963; the mass accumulation rates (MAR) are obtained from the CRS -2 model.**





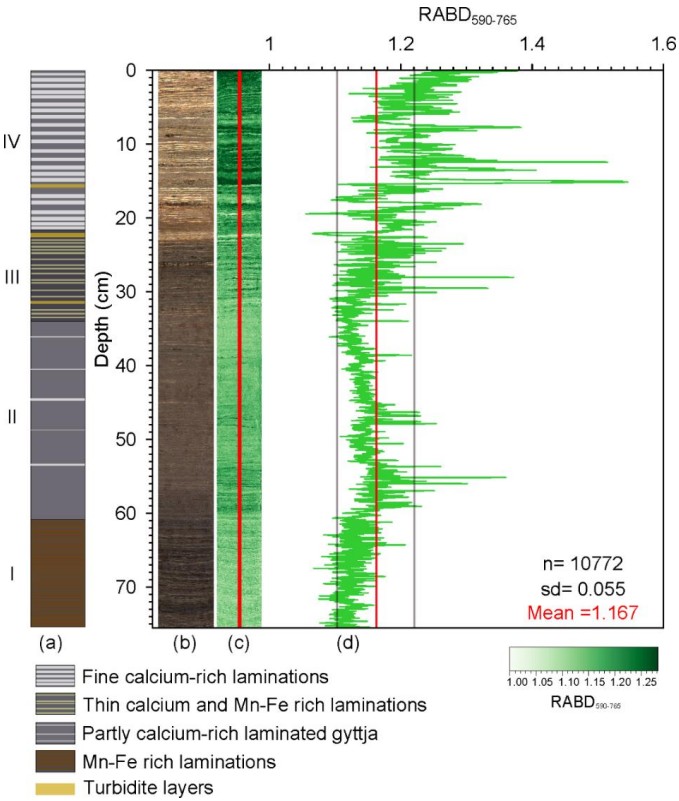


**Figure 3:** (a) Lithological description of Burg17-B sediment core. The intensities of Ca, Fe and Mn in each unit were inferred from XRF-element counts; Yellow colors highlight the turbidite layers identified from the XRF peaks of siliciclastic elements e.g. K, Ti, and Rb. (b) RGB contrast enhanced sediment core picture. (c) The map of the spectral index $RABD_{590-765}$ (i.e. green-pigments) distribution, and (d) the graphic output of $RABD_{590-765}$ spectral index within the boundary of the red lines (c) which shows the 2-mm wide sampling range. The red line in (d) indicates the mean index value and the grey lines represent the one-standard deviations (sd). The colorbar represents the index values of the distribution map (color figure online). n is the number of rows of the $RABD_{590-765}$ index map.

706





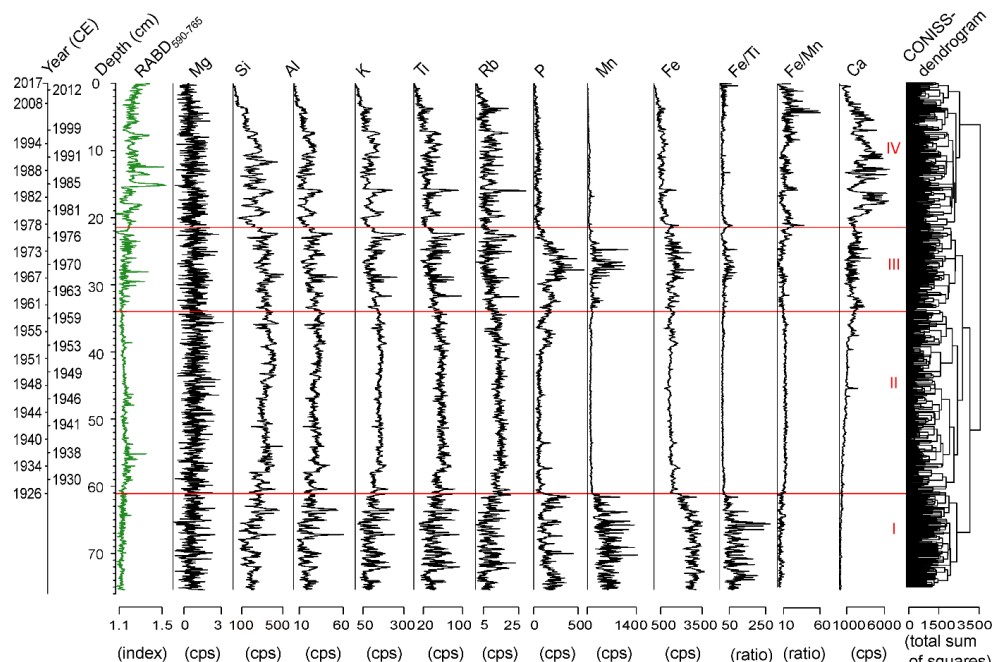

**Figure 4: Stratigraphical records of HSI-inferred green-pigments (RABD$_{590-765}$) and XRF-data in sediments of Core Burg17-B. Elemental counts are represented in cps (counts per second). The red horizontal lines separate the four significant clusters retrieved by the CONISS analysis (color figure online).**



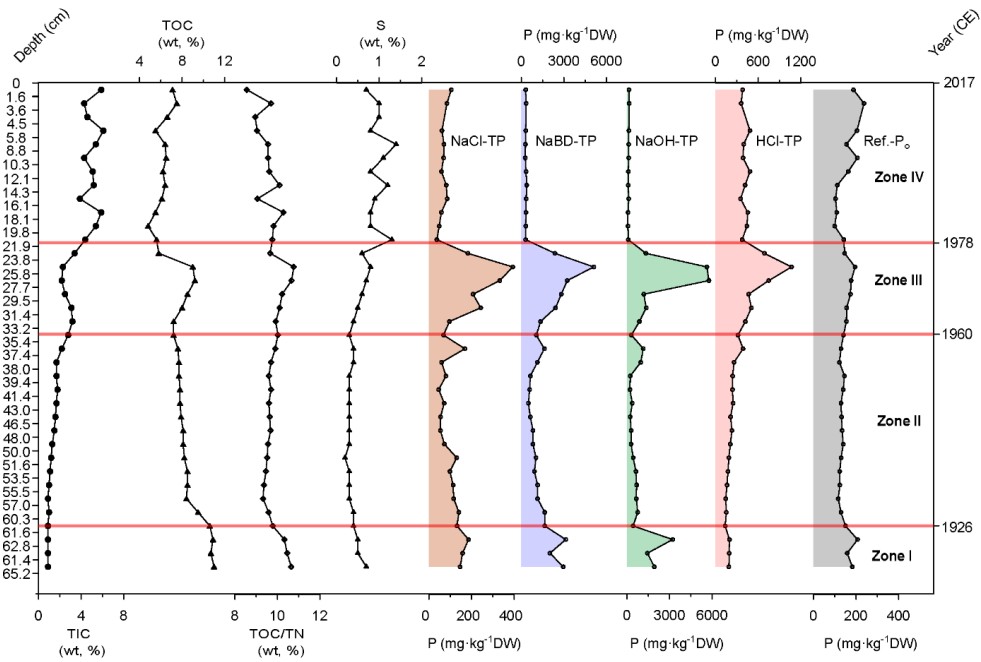

**Figure 5: The stratigraphy of total inorganic carbon (TIC), total organic carbon (TOC), sulfur (S) contents, TOC/TN ratio and five phosphorus fractions in sediments of Lake Burgäschi. The y-axis (left) refers to the sediment depth of Core Burg17-B. The horizontal red lines separate the significant CONISS zones as in Fig. 4. The secondary y-axis (right) indicates approximate ages of sediment inferred from the Burg17-B core chronology.**



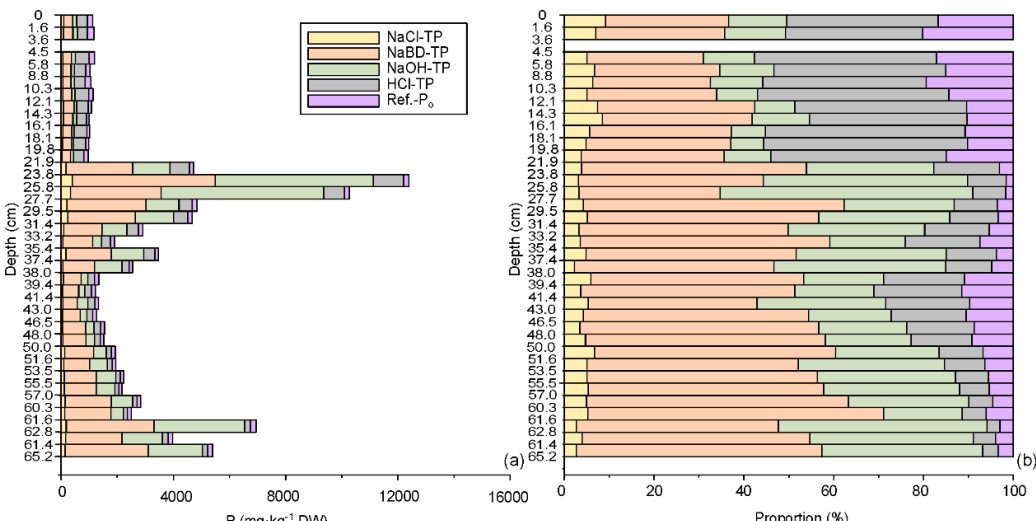

**Figure 6: Vertical profile of (a) P fractions concentrations and (b) their proportions of total P in sediments. The y-axis (left) refers to the sediment depth of Core Burg17-B. Note that the sample between 3.6-4.5 cm depth was removed from dataset because the values were extremely higher than any sample data (data not shown), which is abnormal according to XRF-P counts at the corresponding depth (Fig. S3b). We attributed this to the result of contamination during the sample measurements.**

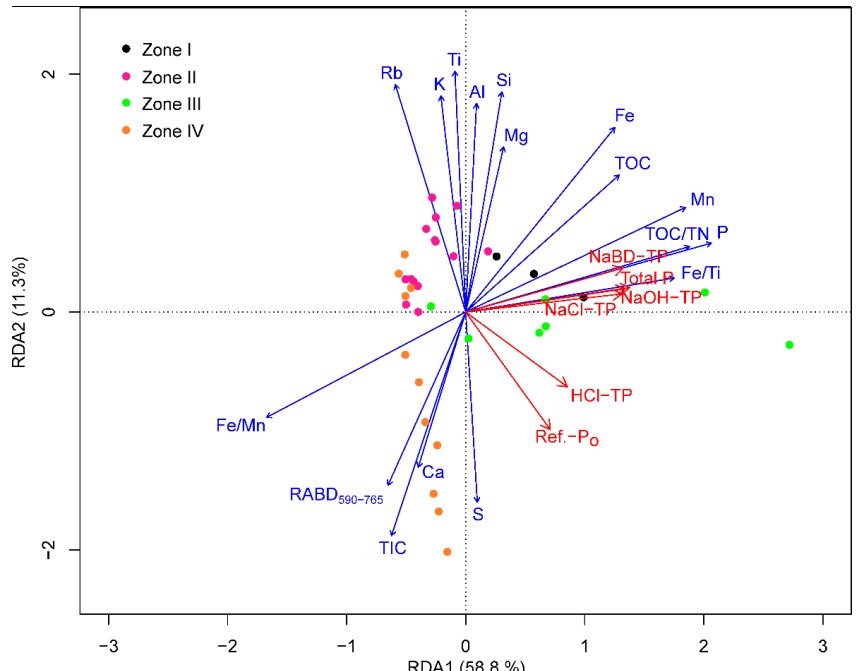



**Figure 7. RDA biplot displaying correlation between response variables (P fraction dataset; red arrows) and**
**explanatory variables (green-pigments and other geochemical records; blue arrows). The colored points represent data**
**points of individual cluster zones.**