# Peer review of "The influences of historic lake trophy and mixing regime changes on long-term phosphorus fractions retention in sediments of deep, eutrophic lakes: a case study from Lake Burgäschi, Switzerland"

_Biogeosciences, 2019_

## Referee Comment (RC1) · Anonymous Referee #1 · 27 Nov 2019

General comments: This is a very careful and detailed state-of-the-art case study of the trophic history of a small Swiss lake (Burgäschisee) exclusively from the analysis of a single sediment core. The authors present an interesting data set that deserves, however, a more deepened discussion and a more complete and careful presentation. More than 50 years ago, lake managers hypothesized based on water column P mass balance considerations that hypolimnetic water withdrawal must result in a decreasing sediment P content. Apart from confirming this prediction, the MS – in its present state

– contributes little to a better understanding of the benthic P cycling. Moreover, I have some reservations in terms of interpretation of in-lake processes and suggest some points to be revised. In fact, the interpretation of the data in the context of sediment diagenesis and processes in the water column (chapter 5.3. onwards) is not consistent. Some statements have no foundations in the results. The discussion contains several contradictions and, therefore, inaccurate conclusions. A wealth of data have been produced from this sediment core but little of it was used to interpret biogeochemical processes in the lake and its sediment. The effects of deep water syphoning starting 1977 is not considered appropriately for the interpretation of sediment profiles and biogeochemical lake processes. The manuscript could be significantly improved and made more attractive for a broader readership if the processes of P scavenging were better characterized (data permitting). The manuscript should be revised aiming to develop a straightforward concept easily explaining benthic P retention as a result of gross P sedimentation, redox dependent benthic transformation and inorganic P sequestration, transient periodic P accumulation in the hypolimnion and P export due to hypolimnetic water withdrawal. Specific points are discussed below.

Specific comments: Lines 65/66: You might wish to replace the sentence "However, the long term influence . . . . . . .. " by the following information: Gächter (1976, Die Tiefenwasserableitung, ein Weg zur Sanierung von Seen. Schweiz. Z. Hydrol. 38: 1-28.) demonstrated that syphoning of hypolimnetic water affects the seasonal lake internal P cycling as follows: 1. P released from sediments during anoxia does not accumulate in the hypolimnion because it is efficiently exported out of the lake, and 2. hence, cannot re-precipitate and settle again to the sediment during spring overturn. 3. Consequently, hypolimnetic withdrawal must result in a phosphorus impoverishment of the sediment and 4. thus, – in the long term – very likely in a decreasing benthic P release during summer stagnation.

Line 165ff: For your further interpretation of the trophic state of the lake it is essential to document whether green pigments preserve well in the sediments over decades. To

my knowledge, the chlorin index (Schubert et al., 2005, Geochem. Geophys. Geosys. 6, 3) that traces chlorophyll and degradation products, increases quite fast downcore (indicating decomposition of chlorophylls).

Line 177: According to your extraction scheme, you determined five operationally defined P fractions (F1 to F4 and the total P). Which fraction represents the refractory organic P? Is it defined as TP – (F1+F2+F3+F4)?

Line 274: "Afterwards" is confusing, because sediment age decreases downwards. Suggestion: Replace "Afterwards" by "then, with decreasing age, they increase . . ... "

Line 280: I suggest to extend Fig. 5 by inserting a profile of the annual total P (TP) retention equaling TP content x MAR and to briefly describe its characteristics here. (see also comment to Lines 311 ff.). For the interpretation of sedimentary processes it might be helpful to present other parameters in amount per area per time (areal mass accumulation rate) as well.

Line 291: Unless it has been shown that "green pigments" behave absolutely refractory (i.e., that they do not degrade with increasing age after deposition), I doubt that it is save to interpret their profile uncritically as an indicator for the lake's productivity at the time when the pigment was buried in the sediment.

Line 292: Delete 'with each other'.

Lines 295 to 297: See comment to line 291

Line 304 ff.: 1943 the lake water level and hence the water table level was artificially lowered by more than 2 m in order to create more crop land (see Guthruf et al, 1999). Discuss the possible effect of this measure on the nutrient load of the lake. It is surprising that this measure is not immediately visible in the sediment profiles and MARs. However, as sedimentation rates are different in the two cores analyzed (Figure 2c), the year 1943 can be located at ∼40 cm or at ∼50 cm sediment depth, which allows quite a range for indicators to look for.

Lines 311 ff.: Fig. 4 provides semi quantitative information about the sediment content of the presented elements that should not be misinterpreted as their more informative annual net-deposition rates (gross deposition minus release). This information could be obtained for TIC and the various phosphorus species by simultaneous consideration of MAR (Fig. 2c) and the information on the corresponding sediment content (Fig. 5).

Line 332-33: I would assume that Fe and Mn in the sediment have their source and continuous supply in the catchment. The pattern is caused by bottom water anoxia, sediment diagenetic processes, and physical mixing processes (given that the deposition of organic matter exceeds the critical threshold. Since laminated Fe/Mn patterns appear already before 1926 I assume that this threshold was exceeded already then, and bottom water anoxia was common during the stratified season?).

Line 355ff: From the Fe/Mn pattern in the sediment one can conclude on the oxygen conditions in the hypolimnion, i.e. physical mixing. It is, however, difficult to conclude on primary productivity. PP attains a maximum (in terms of assimilated carbon) at relatively low phosphorus concentration and does not increase with P ad infinitum. The same is true for $O_2$ consumption in the hypolimnion, which reaches a maximum rate of $\sim$1.1 g $O_2$ m-2d-1 for productive lakes. Moreover, the $O_2$ reservoir of a lake with a small hypolimnion volume as Burgäschisee is used up very quickly, and is thus naturally very sensitive to increasing primary production.

Line 362: Replace "no" by "absent"

Line 373 ff.: According to the P extraction scheme, Ca-bound P would most likely dissolve as HCl-TP (provided that most of the HCl-TP is inorganic P). As the HCl-TP content does not systematically change within Zone IV but MAR decreases with decreasing sediment age, the presented data do not support the conclusion that a changing environment resulted in an increasing rate of benthic Ca-bound P burial. In fact, the annual deposition rate of Ca-bound P [amount per area per time] decreases.

It is not comprehensible that P-coprecipitation with calcite predominates after 1977 "as

an incidence of biologically driven precipitation . . . in highly productive lakes" because the lake was highly productive for decades. My guess is that this clear pattern in the sediment is an effect of the deep water siphoning that was installed at that time, which removed not only hypolimnetic P but also dissolved Fe and Mn, leaving calcite as the main sorbent for P? The installation of the syphoning facility and its effect on the biogeochemistry of the lake must be included in the discussion! It has affected lake processes fundamentally, and the new patterns can only be interpreted when the consequences of the syphoning are considered.

Line 382ff: Replace Âńcoving Âż by Âńcovering" The first sentence is too general as it refers to the whole core. At least in zone IV it is obvious that P retention is mainly due to biogenic calcite formation.

Line 383: The Fe/Ti ratio indicates allochthonous Fe, not autochthonous? In fact, all Fe and Mn originates from the catchment, and the formation of Fe(II) and Mn(II) is a sediment diagenetic process leading to co-precipitation of P during mixing with oxygenated water masses.

Line 383-387: Here you assume that all sediment-P is bound to Fe(III) and Mn(IV) phases. However, at least in zone III, there was a high amount of NaOH-P in addition to NaBD-P indicating that a substantial amount of P was not bound to reducible Fe(III) phases. What is the binding form here? Could it possibly (partly) consist of precipitates of Fe(II)-phosphate or similar? Do you have geochemical indicators to separate Fe(II) from Fe(III) in the sediment?

Line 387-388: ". . . when the hypolimnion had better oxic conditions. . .". What is the base for this assumption? The lakes bottom water during stratification was anoxic already before 1926 (Mn-Fe rich laminations, Figure 3) and productivity even increased afterwards (paragraph line 304, paragraph line 311, paragraph line 346 ('stable anoxic hypolimnetic waters' in zone II), etc.).

Line 393: contradiction: 'Mn and Fe oxyhydroxides' are not 'reduced P-bearing solid

phases'.

Line 394: What are 'more anoxic conditions'? Conditions are either oxic or anoxic.

Line 394-395: The effect of hypolimnetic syphoning must be taken into consideration here.

Line 397-399: '... reduced Mn and Fe preservation suggests ...' Do you suggest that Fe and Mn in zone IV are reduced phases? Why not in other zones? Are there indications that Fe and Mn are present as Fe(II) and Mn(II)?

Line 402: '... decreased P retention was observed'. This reads as if the retention capacity had decreased (and maybe it has, due to much lower concentrations of Fe and Mn). But I assume you mean that the mass accumulation rate of P in the sediment has decreased?

Line 404: change to: " were of autochthonous origin". I do not understand why this is stated. HCl-P might be mainly P scavenged by biogenic calcite precipitation (autochthonous), but what kind of material is the refractory P? By which processes could this be produced?

Line 406: 'Interestingly...'. My guess is that syphoning had an immediate effect on the concentration of TP in the lake. Monitoring data of the water column (unfortunately not shown in your paper) demonstrate a drop of TP during winter mixing from 125 mgP/m3 (1975) to <50 mgP/m3 (1987) (with no data in between). Therefore it is plausible that the P deposition rate decreased steeply.

Line 407: '... the lake in zone IV had higher eutrophic levels... (than in zone III)'. What parameters do you use to characterize trophic state? Certainly not TP, because TP concentrations have decreased 'by a factor of 5-6 since 1978' (line 400).

Line 414: in this context: As TP concentrations have decreased due to syphoning – why should 'the enhanced retention of Ref-P be derived from increased algal refractory organic matter'? I assume that if you calculate the deposition rate of Ref-P (multiplication

with MAR) you would see that it has not changed significantly in recent years – (maybe indicating that it is of allochthonous origin?). In addition, an increasing concentration of any solid organic component towards the sediment surface (younger sediment) may be partly due to still ongoing metabolic processes and thus does not per se indicate an increasing sedimentation rate during more recent years.

Line 417-423: First, the authors do not state any quantitative information on observed water quality parameters. Second, I would speculate that the lake's trophic state did not respond to the hypolimnetic withdrawal because before syphoning P, Fe(II) and Mn(II) released from the sediment accumulated transiently only in the hypolimnion and re-precipitated when Fe(II) and Mn(II) came into contact with O2 without affecting the P load to the trophogenic layer and, hence, primary production. After hypolimnetic withdrawal went into operation, the released P, Fe(II) and Mn(II) are exported, thus, do no longer precipitate and consequently the sediment depletes in P. TP concentrations after winter mixing decreased in subsequent years from 125 mgP/m3 to ∼25 mgP/m3, i.e., the lake was still fully productive. As less Fe and Mn is available to bind P in the sediment since the onset of deep water syphoning, the majority of P ending up in the sediment was scavenged by biogenic calcite precipitation. Briefly, the trophic state of Lake Burgäschi with its high flushing rate (0.7 yr-1) is not strongly affected by the hypolimnetic withdrawal, because in this relatively deep lake, primary production depends primarily on the lake's external but hardly on its internal P-loading. In brief, phytoplankton in the epilimnion "does not care" about hypolimnetic P concentrations.

Line 421: Nitrogen limitation: According to the chemical monitoring data of the water column in the years 2000-2010 concentrations of nitrate were always 0.6 – 3 mgN/L or higher in the trophic zone in September. This excludes nitrogen limitation.

Line 428: This remark insinuates that before 1977 the hypolimnion was partly oxic. This contradicts the observation of laminated Fe/Mn patterns in the sediment even in zone I. In line 347 you interpret Fe/Mn patterns in zone II "as the results of stable anoxic hypolimnetic waters".

Line 431: It is obviously too general and imprecise that 'the dominant factor controlling . . . P in the sediments . . . was found to be autochthonous Fe and Mn content in anoxic sediments . . .' because on line 437 it is stated that '. . . Ca-P fraction predominated in surface sediments after 1977'.

Line 432: Why do you claim that the benthic Fe and Mn was of autochthonous origin? What exactly do you mean?

Line 436-439: Why do you think that hypolimnetic water syphoning can lower the lake's external P loading? This conclusion is not supported by any data.

Line 440: Already discussed above. This is a matter of the algae-available P-load and how much of it can be removed from the lake. Due to the short water residence time, a large fraction of the algae-available P effective in primary production enters the epilimnion of the lake during the stratified period and is not affected by the deep water syphoning. Since the load from the catchment was not at all affected by the installation of the syphon, there is no reason to assume that the P load available for production (i.e. the TP present in the trophic zone after winter turnover plus the load of algae-available P flushed into the lake during the productive period) has significantly decreased to affect primary production. Nitrogen limitation can be excluded, and the effect of global warming might affect the duration of the stratification time but certainly does not significantly increase productivity in Burgäschisee.

Line 443-444: I fully agree here!
* * *

---

## Referee Comment (RC2) · Anonymous Referee #2 · 4 Dec 2019

Luyao Tu and co-authors present a ∼120 year-long sedimentary record from Lake Burgäschi on the Swiss plateau discussing variations in the bottom-water oxygenation state, the trophic state and potential phosphorous retention/release during these varying conditions.

The data set of the study including phosphorous fractions, carbon, sulfur and nitrogen concentrations, XRF core scanning, and hyperspectral imaging is extensive. Yet, in my

opinion it is partly overinterpreted and partly not clearly presented. What I am missing is a better linkage of the data with the development of agriculture and deforestation in the lake's catchment area and with the lake's restoration history. Interpretations on the trophic state and the reconstruction of hypolimnetic oxygenation regimes are given in the discussion section but with only weak links to the human influences that are potentially responsible for these variations. However, since the study cannot contribute much to the already known chemical mechanisms of phosphorous retention/release in lake sediments, it seems important that the findings be interpreted with regard to these same human influences.

Overall, my impression after having read the manuscript was that there is a lot of data, a lot of statistical analysis but the promise from the abstract that I would learn how hypolimnetic anoxia influence lake recovery from eutrophication was not accessible to me. It may be that it is included in the manuscript but then in a form that is difficult to access for the readership.

Accordingly, the manuscript should be revised. More focus should be put on the interpretation of the data in a larger context. But the data should also not be overinterpreted as in section 5.3. Currently, the technical and statistical side dominates. I recommend for instance the design of scenarios in the form of conceptual models (sketches) demonstrating the processes that were dominating the lake during Zones I to IV. For the readership, this would make the outcome of this study much more attractive and accessible.

Specific remarks:

Line 44: 'oxygen levels' (not only oxygen)

More details should be given in the Introduction on the principle and aim of the phosphorous analysis. Also, one sentence more on the technique of the P analysis should be added to '3.4 Phosphorus fractionation scheme and bulk element analyses' where the authors simply refer to Tu et al. (2019) for the details (line 171). This shortness on

the P analysis is questionable given the importance it has for the study. At one point or the other (Introduction or Material and methods) more detail should thus be provided.

Line 90: A lake has always just one single outflow.

Line 91-92: I guess it should read 'the most important lowering'.

Lines 112, 113: Referencing to sections that come later in the manuscript is usually not accepted (to be checked for Biogeosciences).

3.2 Chronology: Great detail on the activity analysis of 137Cs and 210Pb. Might not be necessary, but actually I welcome the point that it is once presented in a manuscript.

Line 152: 'The core surface . . .'

Line 156: Add the appropriate elements to the description for the 10 kV and the 30 kV run, respectively.

Line 187: Provide at least a keyword on the method and not only the naked reference. As a reader I would like to know at least in which direction it goes before deciding if I want to search for the reference.

Line 304: This general increase in sedimentary green pigments I cannot see.

Line 305: 'green-pigment concentrations'

Section 5.3: This section is rather a mix of already presented interpretations on the state of the lake and interpretations that are speculative and are lacking the necessary data foundation (not given by the results from this study).

Line 372: 'Fe contents control'

Lines 438-442: Here, finally a clear statement linking the data with lake restoration, agricultural influences etc. Points like this should be more elaborated during the Discussion section and not only brought forward (a bit out of nowhere) in the Conclusions.

Figure 3 with the lithological interpretations cannot come before the presentation of the

XRF data in Figure 4 as you need the XRF data to define the lithologies.

Figure 4: Too many elements shown. This is a common issue with XRF data. Mg should be deleted, this is noise, Mg is too light to be measured with an Avaatech scanner. Either Al or K is sufficient; I recommend K as it is heavier and therefore presents the more robust result. Ti is sufficient here (delete Rb). You could show Al and Rb in the supplementary material if you wish.

Figure 6: Can be added to Figure 5. Might even be more illustrative to have all the P plots together in one graph.

Figure 7: I do not understand how the colored points of the individual cluster zones are added to this graph. Please explain in the figure caption.

---

## Author Comment (AC1) · 5 Jan 2020

RESPONSE TO COMMENTS of Anonymous Referee #1 Manuscript ID bg-2019-389 titled "The influences of historic lake trophy and mixing regime changes on long-term phosphorus fractions retention in sediments of deep, eutrophic lakes: a case study from Lake Burgäschi, Switzerland".

General response: We would like to express how much we appreciate the comments

and feedbacks provided by Anonymous Referee #1. We have addressed the comments point by point below. As we agree with the comments and have made additional calculations (e.g. fluxes), we are confident that we can address the comments adequately in a revised version of the manuscript.

General comments R1: This is a very careful and detailed state-of-the-art case study of the trophic history of a small Swiss lake (Burgäschisee) exclusively from the analysis of a single sediment core. The authors present an interesting data set that deserves, however, a more deepened discussion and a more complete and careful presentation. More than 50 years ago, lake managers hypothesized based on water column P mass balance considerations that hypolimnetic water withdrawal must result in a decreasing sediment P content. Apart from confirming this prediction, the MS – in its present state – contributes little to a better understanding of the benthic P cycling. Moreover, I have some reservations in terms of interpretation of in-lake processes and suggest some points to be revised. In fact, the interpretation of the data in the context of sediment diagenesis and processes in the water column (chapter 5.3. onwards) is not consistent. Some statements have no foundations in the results. The discussion contains several contradictions and, therefore, inaccurate conclusions. A wealth of data have been produced from this sediment core but little of it was used to interpret biogeochemical processes in the lake and its sediment. The effects of deep water syphoning starting 1977 is not considered appropriately for the interpretation of sediment profiles and biogeochemical lake processes. The manuscript could be significantly improved and made more attractive for a broader readership if the processes of P scavenging were better characterized (data permitting). The manuscript should be revised aiming to develop a straightforward concept easily explaining benthic P retention as a result of gross P sedimentation, redox dependent benthic transformation and inorganic P sequestration, transient periodic P accumulation in the hypolimnion and P export due to hypolimnetic water withdrawal. Specific points are discussed below.

General Response: Reviewer 1 (R1) raises several important questions that help significantly improve the manuscript (see specific comments and responses below). In a revised version, we will implement the following major modifications and emphasize the issues raised by R1: - We will frame our Introduction (and specify/reframe the motivation for your research) more towards the effects in the sediments found after hypolimnetic syphoning. We find it noteworthy (thanks to the suggestion by R1) that the sediments actually do show a positive result (and the one that has been predicted) after the remediation measures, whereas the effects found in the water body (e.g. productivity) remain mixed (BVE/GSA 2007: "30 Jahre Tiefenwasserableitung: Wie geht es dem Burgäschisee heute"). We will also include a discussion about similarities and differences with other lakes where hypolimnetic syphoning has been installed, limnological parameters were monitored in subsequent years (e.g. Lake Mauen, Gächter, 1976, a lake which is larger but shallower). This places our study in a larger context. - We place more emphasis on within-lake processes and the effect of water withdrawal (with regard to Fe and Mn removal from the lake system) to interpret our data from the sediments. - From comments by R1 and literature for Lake Mauen (Switzerland) and lake Burgäschi (our study, Switzerland) it appears that, in Lake Burgäschi, the most obvious improvement of lake remediation (here: syphoning) are actually found in the sediments (net P burial rates, P speciation and potential for the release of internal P loads) whereas relatively little effect is found in the lake itself. Confirming respective predictions (e.g. Gächter, 1976) with sediment data was precisely the reason and motivation for our study. In contrast to well-mixed shallow lakes, there are still very few studies available assessing P speciation and P speciation net burial rates in deep, seasonally or permanently anoxic lakes. Our study is, therefore, important and to some extent unique as we can assess the long-term effects (> 40 years) of lake remediation in a system where also very good limnological monitoring data are available. Accordingly, we modified and shaped the scope of our study (Introduction) to make the new contribution better visible. - Most of the 'contradictions' mentioned by R1 are due to imprecise language. Content-wise, we fully agree. We also point much better to the Supplementary Online Materials (for clarification).

Specific comments: 1. Lines 65/66: You might wish to replace the sentence "However, the long term influence......" by the following information: Gächter (1976, Die Tiefenwasserableitung, ein Weg zur Sanierung von Seen. Schweiz. Z. Hydrol. 38: 1-28.) demonstrated that syphoning of hypolimnetic water affects the seasonal lake internal P cycling as follows: 1. P released from sediments during anoxia does not accumulate in the hypolimnion because it is efficiently exported out of the lake, and 2. hence, cannot re-precipitate and settle again to the sediment during spring overturn. 3. Consequently, hypolimnetic withdrawal must result in a phosphorus impoverishment of the sediment and 4. thus, – in the long term – very likely in a decreasing benthic P release during summer stagnation.

Response: Thank you for pointing to the study by Gächter (1976) in Lake Mauen. In the revised Introduction (revised manuscript), we will summarize the major results of this study (observations from the water body and inferences for the sediment P) for the points 1-4 (mentioned above). Yes, our data (long-term P speciation in sediments, the focus of our manuscript) fully support what has been predicted from limnological observations and mass balance calculations to occur in sediments (e.g. Gächter 1976). This is the most valuable contribution of our manuscript – again, we would like to point out that there are only a very few studies available from deep, seasonally or permanently anoxic eutrophic lakes documenting the long-term effect of hypolimnetic syphoning on P-species net burial rates in sediments (in contrast to studies of P-speciation in shallow holomictic lakes).

2. Line 165ff: For your further interpretation of the trophic state of the lake it is essential to document whether green pigments preserve well in the sediments over decades. To my knowledge, the chlorin index (Schubert et al., 2005, Geochem. Geophys. Geosys.6, 3) that traces chlorophyll and degradation products, increases quite fast downcore (indicating decomposition of chlorophylls).

Response: We have added more interpretation regarding eutrophication and pigment preservation (Line 186-188 in the revised manuscript, Section 3.3). The sediments in

Lake Burgäschi are mostly laminated and organic-rich (Van Raden, 2012), and fully anoxic. In principle, pigments preserve well under such conditions (Reuss, 2005). A recent study on early diagenesis of pigments in laminated anoxic sediments (Rydberg et al., currently under review with JOPL; the sediments are comparable to those of L Burgäschi) suggests (i) that pigment diagenesis is restricted to the topmost sediment (few years) and (ii) that 'unspecific' hyperspectral techniques (similar to our technique) are most robust (much better than e.g. HPLC) to assess 'original' pigment concentrations, because all the pigment degradation products are also summarized. Similarly, our 'green pigment' index (RABD590-765) includes both chlorophylls and all diagenetic products (sum of all), and we do not look at the pigment ratios, 'freshness' or potential for degradation of chlorophylls (cf. Schubert et al., 2005) or Chlorophyll Preservation Index CPI. We add more references (using spectral indices as a proxy for paleoproductivity, e.g. from Lake Ponte Tresa, Southern Switzerland); we are confident that our pigment index does reflect paleoproductivity in Lake Burgäschi (qualitatively, as we use it, not overstating). Reuss, N., 2005. Sediment pigments as biomarkers of environmental change. DMU report.

3. Line 177: According to your extraction scheme, you determined five operationally defined P fractions (F1 to F4 and the total P). Which fraction represents the refractory organic P? Is it defined as TP – (F1+F2+F3+F4)?

Response: Fraction 5 (F5) represents the refractory organic P. TP is the sum of the five fractions (F1-F5). We added the protocol in Fig. S3 (Supplementary Online Materials SOM). We clarify this in the text, see also Fig. S3.

4. Line 274: "Afterwards" is confusing, because sediment age decreases downwards. Suggestion: Replace "Afterwards" by "then, with decreasing age, they increase…. "

Response: Done, modified.

5. Line 280: I suggest to extend Fig. 5 by inserting a profile of the annual total P (TP) retention equaling TP content x MAR and to briefly describe its characteristics here.

(see also comment to Lines 311 ff.). For the interpretation of sedimentary processes it might be helpful to present other parameters in amount per area per time (areal mass accumulation rate) as well.

Response: Thank you for this suggestion. Yes, we calculated and added the fluxes (i.e. Net burial rates; $\mu$gÂůcm-2Âůyr-1) of all P fractions, TIC, TOC TP and sediment mass accumulation rates (MAR) of Core Burg17-C in Fig. S11 (Supplementary online materials). As MARs (calculated from the CRS 210Pb model) are quite constant, results do not change much except for the decreases of HCl-TP flux and Ref.-Po flux after 1978. This is interesting (added in the text).

6. Line 291: Unless it has been shown that "green pigments" behave absolutely refractory (i.e., that they do not degrade with increasing age after deposition), I doubt that it is save to interpret their profile uncritically as an indicator for the lake's productivity at the time when the pigment was buried in the sediment.

Response: See our comment above. Yes, this is a very critical issue which we are aware of. Given existing literature, the sediment taphonomy of L Burgäschi is ideal for pigment preservation. Our record does not show a pronounced 'tailing' in the top 1-3 cm of the sediment which would indicate early diagenetic processes. As mentioned above, the spectral index of "green-pigments" includes all degradation products of chlorophylls. Pigments and their diagenetic products have shown to be a useful reliable proxy for past eutrophication if taphonomic conditions are suitable. We add more references, including Leavitt, P. R and Hodgson DA.: Sedimentary pigments. In: Smol JP, Birks HJB, Last WM (eds) Tracking environmental change using lake sediments, vol 3. Terrestrial, Algal, and Siliceous Indicators. Kluwer, Dordrecht, pp 295–325, 2002.

7. Line 292: Delete 'with each other'.

Response: The correction has been made.

8. Lines 295 to 297: See comment to line 291.

Response: See the responses to Comments #1 and #6.

9. Line 304 ff.: 1943 the lake water level and hence the water table level was artificially lowered by more than 2 m in order to create more crop land (see Guthruf et al, 1999). Discuss the possible effect of this measure on the nutrient load of the lake. It is surprising that this measure is not immediately visible in the sediment profiles and MARs. However, as sedimentation rates are different in the two cores analyzed (Figure 2c), the year 1943 can be located at 40 cm or at 50 cm sediment depth, which allows quite a range for indicators to look for.

Response: It is the CRS-2 model which is valid and places AD 1943 at ca 48 cm sediment depth. At this depth (above and below) none of the sediment proxies or elemental composition does show an anomaly or change. This is interesting indeed. As suggested, more discussion about the effects of lake-water level lowering in 1943 has been added in Line 334-337.

10. Lines 311 ff.: Fig. 4 provides semi quantitative information about the sediment content of the presented elements that should not be misinterpreted as their more informative annual net-deposition rates (gross deposition minus release). This information could be obtained for TIC and the various phosphorus species by simultaneous consideration of MAR (Fig. 2c) and the information on the corresponding sediment content (Fig. 5).

Response: As suggested, we have rewritten this part and more explanation has been given in Line 342-343 (revised manuscript). We have added a Figure with the fluxes for those proxies where this is possible (see above Fig. S11). Calculating fluxes is not possible for all elements measured with XRF on wet sediment (unit cps) because of changes on porosity, water contents among others. Again, MAR and sediment componentry are relatively homogenous throughout the sediment core; thus, we do not expect major differences between concentrations (% or cps) and Flux that would change the interpretation.

11. Line 332-33: I would assume that Fe and Mn in the sediment have their source and continuous supply in the catchment. The pattern is caused by bottom water anoxia, sediment diagenetic processes, and physical mixing processes (given that the deposition of organic matter exceeds the critical threshold. Since laminated Fe/Mn patterns appear already before 1926 I assume that this threshold was exceeded already then, and bottom water anoxia was common during the stratified season?).

Response: Yes, bottom water anoxia in Lake Burgäschi was common during the stratified season, as the lake is quite deep (max. 31m) compared with its small surface 0.21 km2. We refer to van Raden (2012) who described the redox varves in Lake Burgäschi as a result of seasonal changes in redox conditions. Redox varves were present in this lake since prehistoric times. Added in the text.

12. Line 355ff: From the Fe/Mn pattern in the sediment one can conclude on the oxygen conditions in the hypolimnion, i.e. physical mixing. It is, however, difficult to conclude on primary productivity. PP attains a maximum (in terms of assimilated carbon) at relatively low phosphorus concentration and does not increase with P ad infinitum. The same is true for O2 consumption in the hypolimnion, which reaches a maximum rate of 1.1 g O2 m-2d-1 for productive lakes. Moreover, the O2 reservoir of a lake with a small hypolimnion volume as Burgäschisee is used up very quickly, and is thus naturally very sensitive to increasing primary production.

Response: Yes, this is correct and was not clear from the text (wording). We do not draw inferences from the Fe/Mn pattern on PP. The observation of increased PP comes from the pigment data. We precisely meant that oxygenation and the presence of Mn in redox varves is naturally very sensitive to increasing PP. We rephrased and added information from Section 5.1 to discuss lake primary productivity in Zone III (Line 386 revised manuscript).

13. Line 362: Replace "no" by "absent".

Response: Done, modified.

14. Line 373 ff.: According to the P extraction scheme, Ca-bound P would most likely dissolve as HCl-TP (provided that most of the HCl-TP is inorganic P). As the HCl-TP content does not systematically change within Zone IV but MAR decreases with decreasing sediment age, the presented data do not support the conclusion that a changing environment resulted in an increasing rate of benthic Ca-bound P burial. In fact, the annual deposition rate of Ca-bound P [amount per area per time] decreases. It is not comprehensible that P-coprecipitation with calcite predominates after 1977 "as an incidence of biologically driven precipitation : : : in highly productive lakes" because the lake was highly productive for decades. My guess is that this clear pattern in the sediment is an effect of the deep water siphoning that was installed at that time, which removed not only hypolimnetic P but also dissolved Fe and Mn, leaving calcite as the main sorbent for P? The installation of the syphoning facility and its effect on the biogeochemistry of the lake must be included in the discussion! It has affected lake processes fundamentally, and the new patterns can only be interpreted when the consequences of the syphoning are considered.

Response: Thank you for this important notion. Yes, as suggested, we calculated the annual net burial rate of all P fractions and TP in sediments, as shown in Fig. S11 (SOM) or see the Figure above in responses to Comments #5. In fact, this shows that Ca-TP fluxes slightly decrease between ca 1980 and the present. During this time, the Ca-TP fraction evolves to the dominant sedimentary P-fraction (relative to the other P fractions). We also refer to Gächter (1976) who pointed to the relationship between the Fe-P and the Fe-S cycle and discuss the other effects of deep water syphoning (removal of Fe and Mn ions). More interpretation and discussion about the effects of hypolimnetic water withdrawal on annual deposition rates of P fractions since 1977 have been added in Section 5.3.

15. Line 382ff: Replace Â′ncovingÂËŹz by Â′ncovering" The first sentence is too general as it refers to the whole core. At least in zone IV it is obvious that P retention is mainly due to biogenic calcite formation.

Response: Yes, we agree that after 1977, P retention in sediments is mainly due to biogenic calcite formation. This is different from TP in the entire sediment profile which has – prior to hypolimnetic syphoning - a close relationship with autochthonous Fe (Fe/Ti), Mn and hypolimnetic oxygenation (proxy Fe/Mn ratios), as suggested by the results of the RDA analysis. We rephrase the second and third paragraphs in Sect.5.3, and distinguish between Zone IV and the rest (Line 419-423).

16. Line 383: The Fe/Ti ratio indicates allochthonous Fe, not autochthonous? In fact, all Fe and Mn originates from the catchment, and the formation of Fe(II) and Mn(II) is a sediment diagenetic process leading to co-precipitation of P during mixing with oxygenated water masses.

Response: Yes, we agree that all Fe and Mn in sediments originate from the catchments, but in different forms (lithogenic or dissolved in the water). The Fe/Ti ratio represents in fact both, the allochthonous or autochthonous (endogenic and authigenic) Fe sources: assuming that Fe/Ti in lithogenic material is constant, excess Fe/Ti represents the redox-sensitive autochthonous fraction which is relevant for P precipitation (Zarczynski et al., 2019). This is clarified in the revised text. Åżarczyński, M., Wacnik, A. and Tylmann, W.: Tracing lake mixing and oxygenation regime using the Fe/Mn ratio 648 in varved sediments: 2000 year-long record of human-induced changes from Lake Zabinskie (NE Poland), Sci.Total Environ., 657, 585-596, 2019.

17. Line 383-387: Here you assume that all sediment-P is bound to Fe (III) and Mn (IV) phases. However, at least in zone III, there was a high amount of NaOH-P in addition to NaBD-P indicating that a substantial amount of P was not bound to reducible Fe(III) phases. What is the binding form here? Could it possibly (partly) consist of precipitates of Fe(II)-phosphate or similar? Do you have geochemical indicators to separate Fe(II) from Fe(III) in the sediment?

Response: a) We acknowledge that this is important but, unfortunately, we do not have information about the binding form. We also expect diagenetic effects, maybe even

bacterially mediated. b) NaOH-P includes Al/Fe oxyhydroxides bound P which might partly contain Fe (II)-P but we do not have geochemical indicators to separate Fe (II) from Fe (III).

18. Line 387-388: " when the hypolimnion had better oxic conditions: : :". What is the base for this assumption? The lakes bottom water during stratification was anoxic already before 1926 (Mn-Fe rich laminations, Figure 3) and productivity even increased afterwards (paragraph line 304, paragraph line 311, paragraph line 346 ('stable anoxic hypolimnetic waters' in zone II), etc.).

Response: Yes, this was not clear, we meant seasonal anoxia. We changed the sentence into "……better seasonal oxygenation than in Zone II and IV (see Sect. 5.2)". This is based on the presence of redox varves in Zone III.

19. Line 393: contradiction: 'Mn and Fe oxyhydroxides' are not 'reduced P-bearing solid phases'.

Response: Yes, this was misleading. We have changed it into "Mn and Fe minerals".

20. Line 394: What are 'more anoxic conditions'? Conditions are either oxic or anoxic.

Response: We remove "more".

21. Line 394-395: The effect of hypolimnetic syphoning must be taken into consideration here.

Response: Yes, see our comments above. We have considered and added the effects of the hypolimnetic syphoning restoration also in this context.

22. Line 397-399: '…..reduced Mn and Fe preservation suggests…..' Do you suggest that Fe and Mn in zone IV are reduced phases? Why not in other zones? Are there indications that Fe and Mn are present as Fe(II) and Mn(II)?

Response: We apologize for the confusion. We did not mean that "Fe and Mn in zone IV are reduced phases". We have replaced "reduced" by "decreased" to clarify that the

preservation of Fe and Mn in sediments of Zone IV declined.

23. Line 402: '…..decreased P retention was observed'. This reads as if the retention capacity had decreased (and maybe it has, due to much lower concentrations of Fe and Mn). But I assume you mean that the mass accumulation rate of P in the sediment has decreased?

Response: yes, this was meant. We have changed the sentence to be more clear (Line 446-447).

24. Line 404: change to: " were of autochthonous origin". I do not understand why this is stated. HCl-P might be mainly P scavenged by biogenic calcite precipitation (autochthonous), but what kind of material is the refractory P? By which processes could this be produced?

Response: We have added more context: "…as indicated by absent positive correlations between the two fractions and detrital elements such as Ti, K and Al (Fig. 7)" in Line 451-452 to clarify the statement. Refractory P comes mostly from ashing (550°C) of organic matter and, potentially in traces, also from fluorapatite in molasses bedrock of Lake Burgäschi.

25. Line 406: 'Interestingly: : :'. My guess is that syphoning had an immediate effect on the concentration of TP in the lake. Monitoring data of the water column (unfortunately not shown in your paper) demonstrate a drop of TP during winter mixing from 125 mgP/m3 (1975) to <50 mgP/m3 (1987) (with no data in between). Therefore it is plausible that the P deposition rate decreased steeply.

Response: We have rewritten this part and discussed the effect of hypolimnetic withdrawal on Ca-P annual deposition rates. We also have shown hypolimnetic TP monitoring data from 1977 to 2009 in the synthesis figure (i.e. Fig. S10) showing the rapid decrease as the result of hypolimnetic withdrawal.

26. Line 407: '……. the lake in zone IV had higher eutrophic levels……(than in

zone III)'. What parameters do you use to characterize trophic state? Certainly not TP, because TP concentrations have decreased 'by a factor of 5-6 since 1978' (line 400).

Response: "Higher eutrophic levels in Zone IV" was inferred from the interpretation of "green-pigments" index data, and also from the previous studies (see the last paragraph of Section 5.1). We clarified this in the text.

27. Line 414: in this context: As TP concentrations have decreased due to syphoning – why should 'the enhanced retention of Ref-P be derived from increased algal refractory organic matter'? I assume that if you calculate the deposition rate of Ref-P (multiplication with MAR) you would see that it has not changed significantly in recent years – (maybe indicating that it is of allochthonous origin?). In addition, an increasing concentration of any solid organic component towards the sediment surface (younger sediment) may be partly due to still ongoing metabolic processes and thus does not per se indicate an increasing sedimentation rate during more recent years.

Response: Yes, you are right, thank you for this comment. The flux rates revealed a generally weakly decreasing trend of Ref.-Po fraction in recent decades (Figure S10; Supplementary online materials). This goes hand in hand with a decrease in TOC fluxes suggesting a negative trend in aquatic primary productivity. More discussion, as suggested, has been added in the second-last paragraph of Section 5.3.

28. Line 417-423: First, the authors do not state any quantitative information on observed water quality parameters. Second, I would speculate that the lake's trophic state did not respond to the hypolimnetic withdrawal because before syphoning P, Fe(II) and Mn(II) released from the sediment accumulated transiently only in the hypolimnion and re-precipitated when Fe(II) and Mn (II) came into contact with O2 without affecting the P load to the trophogenic layer and, hence, primary production. After hypolimnetic withdrawal went into operation, the released P, Fe(II) and Mn(II) are exported, thus, do no longer precipitate and consequently the sediment depletes in P. TP concentrations after winter mixing decreased in subsequent years from 125 mgP/m3 to _25 mgP/m3,

i.e., the lake was still fully productive. As less Fe and Mn is available to bind P in the sediment since the onset of deep water syphoning, the majority of P ending up in the sediment was scavenged by biogenic calcite precipitation. Briefly, the trophic state of Lake Burgäschi with its high flushing rate (0.7 yr-1) is not strongly affected by the hypolimnetic withdrawal, because in this relatively deep lake, primary production depends primarily on the lake's external but hardly on its internal P-loading. In brief, phytoplankton in the epilimnion "does not care" about hypolimnetic P concentrations.

Response: Yes, limnological data is shown in Suppl. Fig. S10 as mentioned above. Thank you for this comment, which is a very good point for the interpretation, and an important point for discussion in contrast to Lake Mauen (Gächter, 1976) which is larger and shallower. We have added more context and discussion in Line 483-486.

29. Line 421: Nitrogen limitation: According to the chemical monitoring data of the water column in the years 2000-2010 concentrations of nitrate were always 0.6 – 3 mgN/L or higher in the trophic zone in September. This excludes nitrogen limitation.

Response: Agreed. We have removed "nitrogen limitation" from the possible factors.

30. Line 428: This remark insinuates that before 1977 the hypolimnion was partly oxic. This contradicts the observation of laminated Fe/Mn patterns in the sediment even in zone I. In line 347 you interpret Fe/Mn patterns in zone II "as the results of stable anoxic hypolimnetic waters".

Response: We have changed the text in the Conclusions. Although here we have to clarify that "Persistent anoxic conditions in the hypolimnion after ∼1977" is unrelated to the question "whether or not, before 1977, the hypolimnion was partly oxic" (which was seasonally the case).

31. Line 431: It is obviously too general and imprecise that 'the dominant factor controlling P in the sediments.........was found to be autochthonous Fe and Mn content in anoxic sediments.....' because on line 437 it is stated that ':....Ca-P fraction predominated in surface sediments after 1977'.

Response: Yes, this is true, we need to differentiate between the zones. The correction has been made to clarify that "only before 1977, P retention in sediments was mainly controlled by autochthonous Fe and Mn content in anoxic sediments". After 1977, hypolimnetic withdrawal restoration played a role in P retention in sediments.

32. Line 432: Why do you claim that the benthic Fe and Mn was of autochthonous origin? What exactly do you mean?

Response: Autochthonous origin of Fe and Mn means that Fe and Mn preserved in sediments is mainly controlled by redox and diagenetic processes (i.e. in-lake processes) rather than by clastic inputs. That is why we call it "autochthonous". The sentence is clarified.

33. Line 436-439: Why do you think that hypolimnetic water syphoning can lower the lake's external P loading? This conclusion is not supported by any data.

Response: Yes, this is not what we meant. We have corrected the sentence and removed the "external P loading".

34. Line 440: Already discussed above. This is a matter of the algae-available P-load and how much of it can be removed from the lake. Due to the short water residence time, a large fraction of the algae-available P effective in primary production enters the epilimnion of the lake during the stratified period and is not affected by the deep water syphoning. Since the load from the catchment was not at all affected by the installation of the syphon, there is no reason to assume that the P load available for production (i.e. the TP present in the trophic zone after winter turnover plus the load of algae-available P flushed into the lake during the productive period) has significantly decreased to affect primary production. Nitrogen limitation can be excluded, and the effect of global warming might affect the duration of the stratification time but certainly does not significantly increase productivity in Burgäschisee.

Response: Agreed, absolutely. Maybe the argument was confusing, but we did not mean that syphoning would decrease external P loads. We fully agree. This is the reason why it is very important to assess sediment P (fluxes and speciation) in order to evaluate the effect of remediation (in this case hypolimnetic syphoning). We have added more context in the Conclusions to reinforce this implication; moreover and accordingly, we rephrased the Research Questions and Motivation in the Introduction to clarify why it is important to look into sediments.

35. Line 443-444: I fully agree here!

Please also note the supplement to this comment:
https://www.biogeosciences-discuss.net/bg-2019-389/bg-2019-389-AC1-supplement.pdf

―――――――――――――――――――――――――

[Figure]

| Extraction step | Extracted P form | Extracted P fraction |
|---|---|---|

2.5 g wet sediments

NaCl — 0.46 M NaCl, 1h → **Loosely bound P** — NaCl-TP F1

Residue — NaCl rinse →

NaBD — 0.11 M NaBD,1h / pH=7.0 → **Redox-sensitive (Fe, Mn bound) P** — NaBD-TP F2

Residue — NaBD rinse, twice / NaCl rinse →

N₂ atmosphere

NaOH — 0.1 M NaOH, 18h → **Al/non-reducible Fe oxides bound P, and organic P** — NaOH-TP F3

Residue — NaOH rinse / NaCl rinse →

HCl — 0.5 M HCl,1h → **Apatite and other inorganic P** — HCl-TP F4

Residue — NaCl rinse →

Residues — Drying 105°C (24 h), ashing 550°C (2 h) / 1 M HCl, 16 h → **Refractory organic P** — Ref.-P$_o$ F5

**Fig. 1.**

[Figure]

**Fig. 2.**

[Figure]

**Fig. 3.**

---

## Author Comment (AC2) · 5 Jan 2020

RESPONSE TO COMMENTS of Anonymous Referee #2 Manuscript ID bg-2019-389 titled "The influences of historic lake trophy and mixing regime changes on long-term phosphorus fractions retention in sediments of deep, eutrophic lakes: a case study from Lake Burgäschi, Switzerland".

General comments: Luyao Tu and co-authors present a _120 year-long sedimentary

record from Lake Burgäschi on the Swiss plateau discussing variations in the bottom-water oxygenation state, the trophic state and potential phosphorous retention/release during these varying conditions. The data set of the study including phosphorous fractions, carbon, sulfur and nitrogen concentrations, XRF core scanning, and hyperspectral imaging is extensive. Yet, in my opinion it is partly overinterpreted and partly not clearly presented. What I am missing is a better linkage of the data with the development of agriculture and deforestation in the lake's catchment area and with the lake's restoration history. Interpretations on the trophic state and the reconstruction of hypolimnetic oxygenation regimes are given in the discussion section but with only weak links to the human influences that are potentially responsible for these variations. However, since the study cannot contribute much to the already known chemical mechanisms of phosphorous retention/release in lake sediments, it seems important that the findings be interpreted with regard to these same human influences. Overall, my impression after having read the manuscript was that there is a lot of data, a lot of statistical analysis but the promise from the abstract that I would learn how hypolimnetic anoxia influence lake recovery from eutrophication was not accessible to me. It may be that it is included in the manuscript but then in a form that is difficult to access for the readership. Accordingly, the manuscript should be revised. More focus should be put on the interpretation of the data in a larger context. But the data should also not be overinterpreted as in section 5.3. Currently, the technical and statistical side dominates. I recommend for instance the design of scenarios in the form of conceptual models (sketches) demonstrating the processes that were dominating the lake during Zones I to IV. For the readership, this would make the outcome of this study much more attractive and accessible.

Response: We thank the Anonymous Referee #2 for the valuable and thoughtful comments which add substance to our manuscript, balance the interpretation (more cautious) and improve clarity and readability. Some of the are in line with Reviewer 1. Accordingly, we rephrased (clarified) the primary aims of our study, namely to assess and evaluate the lake remediation measurements (here syphoning of hypolimnetic water) from the viewpoint of sediment profiles, i.e. with data on P fractions, P burial rates and risks for internal P release in and from lake sediments. In most cases remediation is assessed with limnological data. Here we can show that the largest effect of remediation is actually observed in the sediment P pool and its fractions; effects on lake productivity are marginal. The view from lake sediments is critical. Indeed, we confirm with sediment data what has been predicted from limnological data. But in contrast to shallow polymictic lakes, there are only a few studies showing sedimentary P fractions and pools for deep seasonally or permanently anoxic lakes. But we agree: our sediment data need much better discussion in the context of the general eutrophication history and hypolimnetic water withdrawal (with processes). Yes, we added a sketch figure (Fig. 8) with the conceptual model for the four stages discussed in the text.

Specific remarks:

1. Line 44: 'oxygen levels' (not only oxygen) More details should be given in the Introduction on the principle and aim of the phosphorous analysis. Also, one sentence more on the technique of the P analysis should be added to '3.4 Phosphorus fractionation scheme and bulk element analyses' where the authors simply refer to Tu et al. (2019) for the details (line 171). This shortness on the P analysis is questionable given the importance it has for the study. At one point or the other (Introduction or Material and methods) more detail should thus be provided.

Response: The correction has been made as suggested. Yes, the Introduction (specifically the Aim and Motivation for our study; the significance of sedimentary P-fraction retention; the research gaps) is rephrased accordingly. We added more text to the P extraction scheme and added a new Figure with the extraction scheme in the Fig. S3 in Supplementary online materials (SOM).

2. Line 90: A lake has always just one single outflow.

Response: Yes, Lake Burgäschi only has one outflow (clarified in new Line 98).

3. Line 91-92: I guess it should read 'the most important lowering'.

Response: We have changed it into "the most recent lowering".

4. Lines 112, 113: Referencing to sections that come later in the manuscript is usually not accepted (to be checked for Biogeosciences).

Response: As suggested, the correction has been made.

5. 3.2 Chronology: Great detail on the activity analysis of 137Cs and 210Pb. Might not be necessary, but actually I welcome the point that it is once presented in a manuscript.

Response: Thank you. Yes, we decide to leave it as is. A proper presentation of the age-depth model is fundamental (and often not made).

6. Line 152: 'The core surface…..'

Response: Done, corrected.

7. Line 156: Add the appropriate elements to the description for the 10 kV and the 30 kV run, respectively.

Response: As suggested, the correction has been made (Line 177-178).

8. Line 187: Provide at least a keyword on the method and not only the naked reference. As a reader I would like to know at least in which direction it goes before deciding if I want to search for the reference.

Response: More description about the method has been added in Line 197 (revised manuscript). Also the extraction scheme is added in Fig. S3 of SOM.

9. Line 304: This general increase in sedimentary green pigments I cannot see.

Response: The slight increase in "green-pigment" index can be seen more clearly from Figure 4d. We have corrected it (Line 332).

10. Line 305: 'green-pigment concentrations'

Response: Done, corrected.

11. Section 5.3: This section is rather a mix of already presented interpretations on the state of the lake and interpretations that are speculative and are lacking the necessary data foundation (not given by the results from this study).

Response: We have re-organized and re-written this section and make the interpretation from our results more logical and clear to follow, meanwhile with support of limnological monitoring data and previous studies.

12. Line 372: 'Fe contents control'

Response: Done, corrected.

13. Lines 438-442: Here, finally a clear statement linking the data with lake restoration, agricultural influences etc. Points like this should be more elaborated during the Discussion section and not only brought forward (a bit out of nowhere) in the Conclusions.

Response: We have added more context in Section 5.1 to explain how the agricultural activities can influence P loads into the lake and lake primary production (Line 337-340). This is a valuable additional comment to Reviewer 1.

14. Figure 3 with the lithological interpretations cannot come before the presentation of the XRF data in Figure 4 as you need the XRF data to define the lithologies.

Response: As suggested, we have changed the order of Figure 3 and 4.

15. Figure 4: Too many elements shown. This is a common issue with XRF data. Mg should be deleted, this is noise, Mg is too light to be measured with an Avaatech scanner. Either Al or K is sufficient; I recommend K as it is heavier and therefore presents the more robust result. Ti is sufficient here (delete Rb). You could show Al and Rb in the supplementary material if you wish.

Response: Yes, we agree. We have removed the XRF elements Al, Rb, Si, Mg and placed them into the new Fig. S6 (SOM).

16. Figure 6: Can be added to Figure 5. Might even be more illustrative to have all the P plots together in one graph.

Response: Yes, we understand the point. But in fact, the two figures illustrate something different: Figure 5 (curve plot) displays the time-series changes of P fractions and LOI data from Zone I to IV. In contrast, Figure 6 (stacked bar plot) mainly aims to show the average proportions of the five P fractions in the sediments. We prefer to leave them as two Figures.

17. Figure 7: I do not understand how the colored points of the individual cluster zones are added to this graph. Please explain in the figure caption.

Response: We clarify the source in the caption of Figure 7.

Please also note the supplement to this comment:
https://www.biogeosciences-discuss.net/bg-2019-389/bg-2019-389-AC2-supplement.pdf

[Figure]

[Figure]

[Figure]

**Fig. 1.**

| Extraction step | | Extracted P form | Extracted P fraction | |
|---|---|---|---|---|

**2.5 g wet sediments**

NaCl — 0.46 M NaCl, 1h → Loosely bound P | NaCl-TP F1

Residue — NaCl rinse →

NaBD — 0.11 M NaBD,1h pH=7.0 → Redox-sensitive (Fe, Mn bound) P | NaBD-TP F2

Residue — NaBD rinse, twice / NaCl rinse →

N₂ atmosphere

NaOH — 0.1 M NaOH, 18h → Al/non-reducible Fe oxides bound P, and organic P | NaOH-TP F3

Residue — NaOH rinse / NaCl rinse →

HCl — 0.5 M HCl,1h → Apatite and other inorganic P | HCl-TP F4

Residue — NaCl rinse →

Residues — Drying 105°C (24 h), ashing 550°C (2 h) / 1 M HCl, 16 h → Refractory organic P | Ref.-Pₒ F5

**Fig. 2.**

---

## Author Response (AR2)

Manuscript ID bg-2019-389 titled "**The influences of historic lake trophy and mixing regime changes on long-term phosphorus fractions retention in sediments of deep, eutrophic lakes: a case study from Lake Burgäschi, Switzerland**".

We would like to thank the reviewer for the helpful comments to improve the quality of this manuscript. Our response to the reviewer follows point by point.

**Reviewer concerns:**
The authors have appropriately addressed all comments and suggestions. I add some minor suggestions that may help improving the text:

1. Line 13: add "… on short time scales in shallow lakes"
   **Response:** Done, modified.

2. Line 14: replace "eutrophication" by "trophic state"
   **Response:** Done, modified.

3. Line 15: replace "reduce" by diminish"
   **Response:** Done, modified.

4. Line 18-19: replace "…provides information…" by "…provide information on the benthic P retention under the influence…".
   **Response:** Done, modified.

5. Line 21: delete "restoration". Delete "the operation of"
   **Response:** The two modifications are done.

6. Line 25: replace "hypolimnetic had better…" by "…sediment overlaying water was seasonally oxic"
   **Response:** Done, replaced.

7. Line 26-27: I suggest: "…in the hypolimnion and due to hypolimnetic water withdrawal increasing the P export out of the lake, net burial rates of total and labile P fractions decreased considerably in surface sediments."

**Response:** As suggested, the sentence has been modified.

8. Abstract: As you mention "mixing regime" in the title you should explain in the abstract if and how hypolimnetic withdrawal affected the general character of the mixing regime.

   **Response:** We have added the sentence "Also, this restoration has not enhanced water-column mixing and oxygenation in hypolimnetic waters." (Line 33) to indicate that hypolimnetic withdrawal did not improve the lake vertical mixing.

9. Line 88-89: Omit sentence

   **Response:** Done, modified.

10. Line 93: replace "has operated in the lake" by "is in operation"

    **Response:** Done, modified.

11. Line 103: replace "agricultural lands" by "farmland"

    **Response:** Done, modified.

12. Line 106: "The mean annual AIR temperature…"

    **Response:** Done, it is corrected.

13. Line 115: replace "algae" by "algal".

    **Response:** Done, modified.

14. Line 115: What does "oxygenation condition" mean in this context?

    **Response:** We have replaced "oxygenation" into "oxic/anoxic".

15. Line 128: Does "afterwards" mean that bulk elements were determined in the remains of the sediment samples after the P extraction procedure? If not, please clarify!

    **Response:** We apologize for the confusion. We have clarified the texts by replacing the sentence with" After sampling for P extraction, the remaining sediment of the 2-cm slice was………". (Line 143)

16. Line 287: delete "relatively".

    **Response:** Done, modified.

17. Line 288: replace "during" by ""within"

    **Response:** Done, modified.

18. Line 289: delete "reduced"

    **Response:** We have replaced "reduced" with "decreased" to clarify that the labile P fractions and TP concentrations were decreased in Zone II.

19. Line 316: replace "had" by "received"

    **Response:** Done, modified.

20. Line 319-320: "…Lake Burgäschi was likely oligo- to mesotrophic."

    **Response:** Done, modified.

21. Line 331: replace "and" by "indicating".

    **Response:** Done, modified.

22. Line 332-333: delete "reveal strong positive trends in lake eutrophic levels. The significant eutrophication in Lake Burgäschi might have caused"

    **Response:** Done, modified.

23. Line 334: delete "many"

    **Response:** Done, modified.

24. Line 350: replace "during" by "in zones I to III, Mn and Fe varied mostly independent…""

    **Response:** Done, modified.

25. Line 353: Specify: What sort of ratios?

    **Response:** It is the ratios of Fe/Mn from XRF measurements. We have clarified it by adding "The proxy of….". (Line 383-384)

26. Line 386: delete "completely"

    **Response:** Done, modified.

27. Line 387: replace "circulation" by "overturn". Delete "still"

    **Response:** Done, modified.

28. Line 414: are retention and NBR not synonyms?

    **Response:** They are not synonyms because they have different units. In the context, "retention" refers to the contents or the concentrations (mg/kg); but NBR is the net flux with the unit of $\mu g/(cm^2 \cdot yr)$.

29. Line 457: If the epilimnion P concentrations exceed about 10 µg/l phytoplankton growth is likely not P limited.

    **Response:** Yes, the total P data in hypolimnion of Burgäschisee is between 10-50 µg/L (see Figure S11 in Supplementary Online Materials) but the epilimnion is expected to have lower P concentrations, probably < 10 µg/L sometimes.

    According to the study of GBL (1995), the algae-available ortho-phosphate was almost completely used up in the epilimnion during growth season and the high nitrogen concentrations (0.5-3 mg/L) were measured at the same time. We are assuming that phytoplankton production in Burgäschisee is likely limited by phosphorus at least during growth season. We have clarified it in Line 488-490.

    *GBL. Burgäschisee. Resultate der Wasser- und Planktonuntersuchungen 1977-1995. Office for Water Protection and Waste Management of the Canton of Bern, 1995.*

30. Line 460: "… the high lake external P load into the epilimnion…"

    **Response:** Done, modified.

[revised manuscript text omitted]